# Set-LLM: A Permutation-Invariant LLM

**Beni Egressy**[1]    **Jan Stühmer**[1,2]

[1]Heidelberg Institute for Theoretical Studies
[2]IAR, Karlsruhe Institute of Technology

## Abstract

While large language models (LLMs) demonstrate impressive capabilities across numerous applications, their robustness remains a critical concern. This paper is motivated by a specific vulnerability: the order sensitivity of LLMs. This vulnerability manifests itself as the order bias observed when LLMs decide between possible options (for example, a preference for the first option) and the tendency of LLMs to provide different answers when options are reordered. The use cases for this scenario extend beyond the classical case of multiple-choice question answering to the use of LLMs for multidocument tasks and as automated evaluators in AI pipelines. We introduce Set-LLM, a novel architectural adaptation for pretrained LLMs that enables the processing of mixed set-text inputs with permutation invariance guarantees. The adaptations involve a new attention mask and new positional encodings specifically designed for sets. We provide a theoretical proof of invariance and demonstrate through experiments that Set-LLM can be trained effectively, achieving comparable or improved performance and maintaining the runtime of the original model, while altogether eliminating order sensitivity.

## 1 Introduction

The remarkable achievements of Large Language Models (LLMs) in recent years [1, 11, 16] have propelled their adoption across a wide range of applications, including safety-critical and sensitive domains such as medicine and finance [25, 54]. As such, the eye-catching drops in performance from adversarial attacks can be all the more alarming [13, 38]. One such attack, shown in Figure 1, is as trivial as permuting the choices in multiple-choice questions, which Zong et al. [57] demonstrate can degrade an LLM's performance from "good" to worse than random.

This sensitivity to input order becomes even more critical given the increasing reliance on LLMs to compare and evaluate the output of other LLMs [8, 18, 45, 53]. Indeed, LLM-as-a-judge is widely used as an evaluation metric [2, 53], and is also used to annotate LLM-generated output to create new data sets and to decide between possible reasoning paths when solving complex problems [2, 10, 12, 26]. This inherent order sensitivity directly undermines the reliability of these pipelines.

We propose Set-LLM[1], a permutation-invariant LLM architecture that eliminates this problem entirely. Set-LLM guarantees consistent responses by building permutation invariance directly into the model architecture. Moreover, it achieves these guarantees while retaining or improving performance on set-input tasks.

Set-LLM is based on the well-known observation that the attention mechanism underpinning all of the recent LLM architectures is permutation invariant by construction. In fact, to force them to take into account the order of the input tokens, almost all models use some form of positional encoding [37, 39, 43]. However, Kazemnejad et al. [17] prove that even without positional encodings, the causal attention mask used in decoder-only LLMs is sufficient to completely reconstruct the input order. We therefore remove positional encoding and causal masks as the first steps towards Set-LLM.

---

[1]All code is available under open licenses at `https://github.com/hits-mli/set-llm`.

39th Conference on Neural Information Processing Systems (NeurIPS 2025).

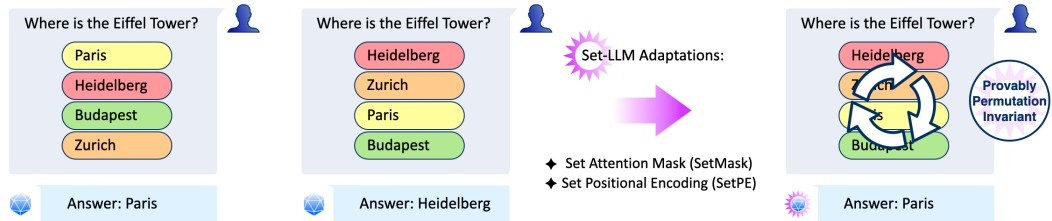

Figure 1: An example of the vulnerability of LLMs to choice permutations. The LLM's response changes simply due to a reordering of the answer options. (Example for illustrative purposes only.) Set-LLM eliminates this vulnerability by building invariance directly into the model architecture.

Our complete approach comprises four steps: (1) Removing sequential position encoding, (2) Removing the causal mask, (3) Adding permutation-invariant set position encoding (SetPE), and (4) Adding a permutation-invariant set attention mask (SetMask). These steps are illustrated in Figure 2. Together, the first two steps result in a *bag-of-words* (BoW) model, i.e., a model with no information about the token order, and the third and fourth steps add the order information we want back in.

We prove that Set-LLM is permutation-invariant, meaning it is guaranteed to give the exact same output whatever input order is chosen. Moreover, we demonstrate that the Set-LLM adaptations can be combined with different decoder-only LLMs and do not depend on specific architectures or model versions, which can quickly become outdated. We run experiments with five different base models on four multiple-choice datasets and two multi-document use cases to show the approach's versatility.

We summarize the main contributions as follows:

- We propose Set-LLM, the first permutation-invariant decoder-only LLM.
- We prove Set-LLM guarantees robustness to permutations, eliminating order sensitivity.
- We demonstrate that Set-LLM can be trained effectively with different base LLMs, consistently matching or outperforming the base models with random-order inputs, and significantly outperforming them with adversarial-order inputs.
- This enables a more robust and efficient evaluation framework for multiple-choice question answering and for the use of LLMs in multi-document applications.

## 2  Background: transformers and positional encoding

The Set-LLM adaptations involve changes to the attention mask and positional encoding of transformer-based LLMs. We first describe these components before introducing Set-LLM.

### 2.1  Attention scores

Most state-of-the-art LLMs are based on the transformer architecture, made up of multiple attention layers stacked on top of one another [43]. Given the $d$-dimensional representation $X \in \mathbb{R}^{N \times d}$ of $N$ tokens, raw (unnormalized) attention scores (or weights) are calculated as

$$Z = \text{attn}(X, X, W_Q, W_K, d_K) = XW_Q(XW_K)^T / \sqrt{d_K}, \tag{1}$$

where $W_Q, W_K \in \mathbb{R}^{d \times d_k}$ are the query and weight matrices, respectively, and $d_K$ is a scaling factor often chosen to be the dimension of the keys.

In a causal transformer, attention scores are masked to ensure that tokens can only *attend* to preceding tokens, before being normalized through a softmax layer. Masking is usually represented by a matrix of 1's and 0's, $M \in \{0, 1\}^{N \times N}$, where $M_{ij} = 1$ if token $i$ can *attend* to token $j$. However, masking can also be denoted by a directed graph $G^M = (V, E)$, where $(j, i) \in E(G^M) \iff M_{ij} = 1$, i.e., if information flows from token $j$ to token $i$. So for a causal mask, $(j, i) \in E(G^M)$ if and only if $j \leq i$. Let $\mathcal{N}_i = \{j \mid (j, i) \in E(G^M)\}$ denote the *neighborhood* (or *field of view*) of the token $i$. Then we can write the normalized attention weights as

$$A_{ij} = \text{softmax}_{G^M}(Z_{ij}) = \frac{\exp(Z_{ij})}{\sum_{k \in \mathcal{N}_i} \exp(Z_{ik})} = \frac{exp(Z_{ij})}{\sum_k M_{ik} exp(Z_{ik})}, \tag{2}$$

if $(j, i) \in E(G^M)$, and $A_{ij} = 0$ otherwise.

LLMs all use some kind of masking. Decoder-only architectures use a causal attention mask, whereas encoder-decoder architectures, such as T5 [34], used bidirectional (or fully-connected) attention for the prefix (or prompt) and causal attention for the output (or response). We refer to this as *prefix masking*. Figure 2 illustrates the causal and prefix masks in both the matrix and graph forms.

Given normalized attention weights, the attention layer is completed by taking weighted averages of the token neighborhoods:

$$X^{(t+1)} = A^{(t)} X^{(t)} W_V, \tag{3}$$

where $W_V$ is the value matrix, and the superscript indicates the model layer.

## 2.2 Positional encoding

In addition to the causal mask, transformers use positional encoding to introduce positional information. There are two common variants: *absolute positional encodings* and *relative positional encodings*. Absolute positional encodings assign consecutive integers to the tokens, starting at 0. These are usually embedded with an encoder layer and concatenated with the corresponding input token embeddings. The neural network can use these embeddings to "understand" word order. The attention scores depend on the token embeddings $X_i, X_j$ and the absolute positions $i$ and $j$:

$$Z_{ij} = \text{attn}_{abs}(X_i, X_j, W_Q, W_K, d_K, i, j). \tag{4}$$

The positional information is often incorporated into the token embeddings, so row $i$ of $X$ is $x_i = \psi(q_i, i)$ for some function $\psi : \mathcal{T} \times \mathbb{N} \to \mathbb{R}^d$, and there is no further dependence on $i$, reducing $\text{attn}_{abs}$ to Equation (1). We formulate our proofs using this notation.

On the other hand, relative positional encodings use the relative distance $(i - j)$ of tokens in attention calculations. The attention scores depend on the token embeddings $X_i$, $X_j$, and the relative distance:

$$Z_{ij} = \text{attn}_{rel}(X_i, X_j, W_Q, W_K, d_K, i - j). \tag{5}$$

In this sense, relative position encodings can be seen as a special case of absolute position encodings, where a translation symmetry on positions is enforced, i.e. shifting the absolute positions by $m$ does not change the attention scores and therefore the attention layer outputs:

$$\text{attn}(X_i, X_j, W_Q, W_K, d_K, i - j) = \text{attn}(X_i, X_j, W_Q, W_K, d_K, (i + m) - (j + m)). \tag{6}$$

## 3 Methods: Set-LLM

LLMs are used for a variety of tasks, and many of them have sets in the input instructions. This includes answering multiple-choice questions and comparing LLM-generated outputs. For example, [{"Which city is the capital of France:"}, {"Budapest", "Paris", "Heidelberg", "Zurich"}].

More generally, if $\mathcal{T}$ denotes the token vocabulary, then a mixed set-text instruction can be written as $q = [q_0, q_1, \ldots, q_n]$, where each $q_i$ is a set of token sequences: $q_i = \{s_0, s_1, \ldots, s_{n_i}\}$ and $s_j = [\tau_0, \tau_1, \ldots, \tau_{n_{i,j}}]$, with all $\tau_i \in \mathcal{T}$. If $q = [q_0]$ and $|q_0| = 1$, then we are back to the regular case where an instruction is a single sequence of tokens. In the above example, depending on the tokenization, we might have $q_0 = \{["Which", " city", \ldots]\}$ and $q_1 = \{["Buda", "pest"], ["Paris"], \ldots\}$.

For ease of notation, we use global indexing, where tokens in a set of choices are also numbered consecutively (in the default order in which the set is provided). Then the tokens of $q$ are $[t_0, t_1, \ldots, t_N]$, with $t_i \in \mathcal{T}$, where $N$ is the total number of tokens in $q$. We use $s(t_i)$ to denote the token sequence containing the token $t_i$ and $q(t_i)$ to denote the set containing $s(t_i)$.

Since LLMs take sequences as input, one would typically force an order onto sets within mixed input. However, ideally, we would like an LLM, whose output does not depend on this order. Our proposed approach, Set-LLM, achieves this in four steps: (1) Remove sequential position encoding, that is, set all positions to 0 (also called NoPE – No Positional Encoding); (2) Remove the causal mask and replace it with a prefix mask; (3) Add permutation-invariant set position encoding (SetPE); and (4) Add permutation-invariant set attention masking (SetMask). The steps are illustrated in Figure 2.

The first two steps already guarantee set permutation invariance. In fact, they create a bag-of-words (BoW) model that ignores the order of all input tokens, not just the order of elements within a set. Clearly, BoW models have their limitations, since word order is critical to language. Steps (3) and (4) are therefore crucial in reintroducing the order information within the different spans of text.

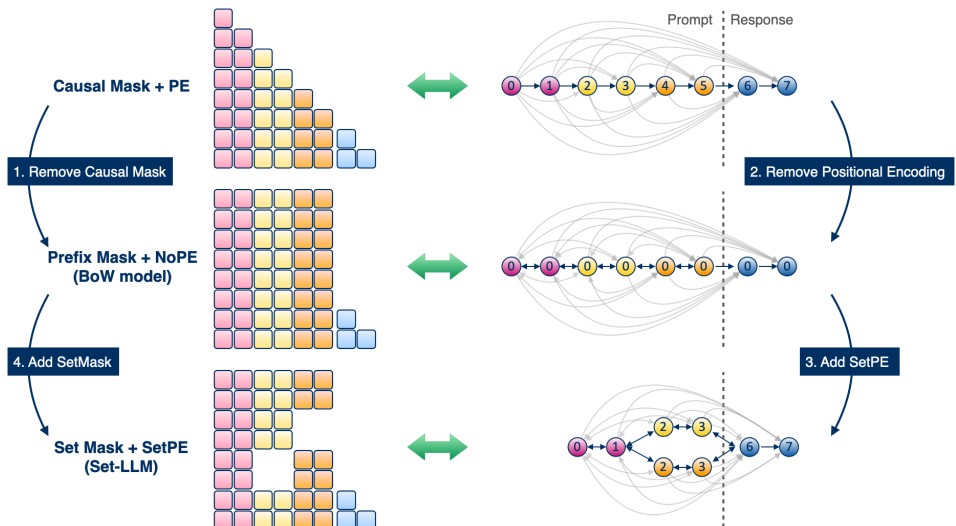

Figure 2: Three types of attention masks and their corresponding directed graphs. The colored squares on the left indicate attention scores that are not masked. For example, in a causal mask, the $4^{\text{th}}$ token attends to the first 4 tokens, and the remaining tokens are masked. The circles on the right represent the tokens as nodes of an attention graph. Red, orange, and yellow tokens correspond to the prompt, and blue tokens correspond to the response. Orange and yellow tokens corresponds to elements of a set. Causal masks are standard in decoder-only LMs, whereas prefix masks are used in bidirectional encoder-decoder LMs. SetMask is introduced in this work. Some edges are grayed out and self-loops are omitted to improve clarity. In addition, the figure shows three types of token positions, standard consecutive positions (PE), *no positional encoding* (NoPE), and *set position encoding* (SetPE). These are indicated by the numbers inside the token nodes on the right.

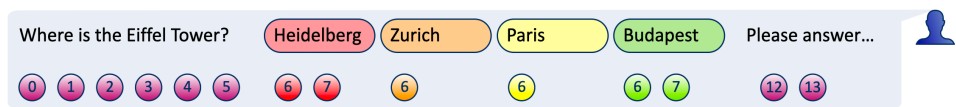

Figure 3: An example of a multiple-choice question with set positional encoding (SetPE) positions.

## 3.1 Set position encoding (SetPE)

While BoW models inherently disregard the order of set elements, they also disregard the word order within the text. To overcome this limitation, we introduce set position encoding (SetPE). For SetPE, we generate *SetPE positions* and use them to calculate absolute or relative positional encodings. The idea is to number tokens consecutively, but to number elements of a set from the same starting position. An example is provided in Figure 3, along with pseudocode in Section B.

The example shows how SetPE positions align with standard absolute positions for "regular" text (starting at 0). However, when a set of options appears (e.g., at position 6), all options within that set are numbered consecutively from that starting position. This ensures that no order is forced upon the options, but the token order within the options is clear. Positions resume with their absolute positions after a set of options (continuing with 12 in the example). We denote SetPE positions by the function $\phi$ (or $\texttt{pos}$ in the pseudocode), i.e., the position of the $i^{\text{th}}$ token of query $q$ is written $\phi(q)|_i = \texttt{pos}[i]$.

Given the SetPE positions, we can calculate absolute or relative positional embeddings. When using absolute positional encoding, the SetPE positions simply replace the absolute positions and are encoded and concatenated with the input token embeddings. When using relative positional encoding, the difference between SetPE positions is used to calculate relative positional embeddings rather than the difference between absolute positions. All LLMs in this paper specifically use RoPE [39]. In this case, the SetPE positions determine the angles of rotation for the token embeddings.

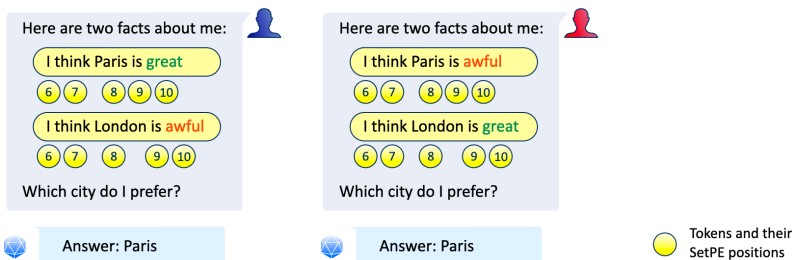

Figure 4: A failure case for an LLM with set position encoding (SetPE) but without set attention mask (SetMask). SetPE positions are shown for the tokens that are part of the set (of facts). Without the SetMask, the model is unable to distinguish the two inputs. The model can not "know" which position 8 token belongs to which position 10 token.

## 3.2 Set attention mask (SetMask)

When using prefix mask, SetPE is insufficient for distinguishing mixed inputs. Consider the example of two inputs in Figure 4, which contain two sets of opposing facts. Clearly, one person prefers Paris, while the other prefers London, but an LLM with prefix masking and SetPE will output the same next token distributions. To see this, note that all tokens in the two cases receive the same SetPE positions, although *great* and *awful* have been switched. Since the prefix mask is fully connected, the attention layer outputs are then identical up to switching the two respective embeddings.

To address this shortcoming, we introduce set attention masking (SetMask). The idea is to use the attention mask to distinguish between the two cases in Figure 4. SetMask is constructed by starting with a prefix mask and removing all edges between tokens of different elements of the same set (that is, setting the respective matrix entries to $0$). An illustration in both matrix and graph forms can be seen in Figure 2 (bottom). More precisely,

$$M_{ij} = \begin{cases} 0 & \text{if } q(t_i) = q(t_j) \text{ and } s(t_i) \neq s(t_j) \\ 1 & \text{else.} \end{cases} \tag{7}$$

Since generation is autoregressive, response tokens can only attend to prompt tokens and preceding response tokens. SetMask is therefore extended to the response in the same way as causal mask and prefix mask. With SetMask, the tokens corresponding to *great* and *awful* in Figure 4 have different neighborhoods in the two inputs, leading to different embeddings and different next token distributions.

## 3.3 Permutation invariance

We claim that by construction, an attention layer with SetPE and SetMask is set permutation equivariant. Then, since all intermediate layers are set permutation equivariant and the final layer of an LLM is permutation invariant, it follows that the whole network is set permutation invariant [4].

**Theorem 1** (Set Permutation Equivariance). *Let $\pi$ be a permutation corresponding to permuting elements of sets in a mixed set-text input, and let $P$ be the corresponding permutation matrix. If $X^{(t+1)} = A^{(t)}X^{(t)}W_V$ is the output of an attention layer with SetPE, SetMask, and absolute positional encoding, then $\tilde{X}^{(t+1)} = PX^{(t+1)}$. In other words, attention with SetPE and SetMask is equivariant to set permutations of mixed set-text input.*

**Theorem 2.** *Attention with SetPE and SetMask reduces to attention with PE and prefix masking when the input is a single sequence of tokens (i.e., when the input does not contain sets).*

Both proofs can be found in Section C.

## 4 Experimental setup

To evaluate Set-LLM, we test it with $5$ models on $4$ multiple-choice datasets. Multiple-choice questions are the ideal testbed for Set-LLM since there are widely-available, well-established benchmarks.

**Datasets.** Models are evaluated on four popular multiple-choice reasoning benchmarks: PIQA [3], ARC-Challenge [5], CommonsenseQA [40], and SIQA [36]. Each task consists of a series of questions, each with multiple choices, where only one answer is correct. The number of choices per question varies by the dataset. In the original setup, PIQA, ARC-Challenge, and CommonsenseQA provide only the question in the prompt, and the different answer choices are run through the model to test them as continuations. The choice with the highest log-likelihood is selected as the answer.

To adapt these datasets to our research setting, we modify the prompt so that the choices are provided as part of the question, and the order of these choices can be permuted. Finally, in the case of SIQA, although the original prompt includes the choices, we remove the numbering since this implicitly assigns an order to the options and breaks the permutation invariance. All original and modified prompts can be found in Section D.3. We use the standard train-evaluation splits for all benchmarks.

**Additional pretraining data.** The Set-LLM adaptations fundamentally change the input of the model and the inner attention mechanism, so the adapted models require training to function in their new setups. To help the models, we experiment with additional pretraining. We use a high-quality subset (approximately 10k examples) of the cleaned UltraFeedback instruction-following dataset [7], attained by following the data preprocessing steps in [21]. We use $^{\text{Ultra}}$ in the results to indicate additional pretraining. More information on instruction-finetuning can be found in Section D.1.

**Evaluation modes.** The experiments involve two evaluation modes: **(1) Random Order:** For each input question, we test all[2] permutations of the answer choices, and calculate the average accuracy, **(2) Adversarial Order:** For each input question, we test all[2] permutations of the answer choices and use a permutation where the LLM returns a wrong answer, if one exists. Accuracy is then the proportion of questions for which the LLM remains correct across all possible permutations.

**Base models.** We evaluate the proposed method using several popular pretrained decoder-only language models: Gemma 2B and 7B [42], Llama 3.2 1B, Llama 3.2 3B, and Llama 3.1 8B [11].[3] We select these models to test different architectures and model sizes.

**Baselines.** We consider the following baselines: (1&2) A pretrained and a finetuned LM using the original dataset prompts (Causal Mask+PE* Pretrained & Causal Mask+PE* Finetuned), (3&4) A pretrained and a finetuned LM using the modified dataset prompts (Causal Mask+PE Pretrained & Causal Mask+PE Finetuned), (5) A finetuned LM with additional pretraining on UltraFeedback [7] using the modified dataset prompts (Causal Mask+PE$^{\text{Ultra}}$ Finetuned).

An alternative to a permutation-invariant architecture is to run all possible permutations of the input through the LLM and pick the option with the most "votes".[4] This approach, **majority vote** [57], can make any model permutation-invariant. However, it also comes with an exponential factor runtime overhead, since the model has to be run $k!$ times, where $k$ is the number of options. We include a majority vote for baselines 3, 4, and 5 from above. These models use the modified prompt containing the answer choices, which can be permuted to calculate the majority vote. Finally, we include another permutation-invariant baseline: **no options**. Here, the questions are asked without providing the answer options. We finetune a Causal Mask+PE$^{\text{Ultra}}$ model on each dataset.

**Training setup.** We update the model weights using LoRA [15] applied to all linear layers of the multilayer perceptron (MLP) and self-attention layers. Details about the hyperparameter settings can be found in Section D.4. We finetune models separately on each benchmark. We train all models on a single Nvidia H200 GPU with training times ranging between 1 and 4 hours for one model on one benchmark. A comparison of baseline and Set-LLM runtimes is provided in Section E.7.

We train all our models with bfloat16 precision. However, we use full 32-bit floating point precision for all evaluation runs. This proves crucial in ensuring permutation invariance in practice. Although the Set-LLM architecture is provably permutation-invariant (Theorem 1), permuting the input tokens can lead to a different order of the low-level computations resulting in minor inconsistencies, which add up layer by layer. We do not observe any inconsistencies using 32-bit floating-point precision.

---

[2]For CommonsenseQA, only the first $24$ (of a possible $5! = 120$) permutations are tested.

[3]google/gemma-2b, google/gemma-7b, meta-llama/Llama-3.2-1B-Instruct, meta-llama/Llama-3.2-3B-Instruct, meta-llama/Llama-3.1-8B-Instruct [46, Huggingface]

[4]In the event of a tie, the winner is chosen uniformly at random from the top-voted options.

Table 1: Gemma 2B baselines on four multiple-choice datasets. All scores are accuracies (%). *Results using the original dataset prompts, which for PIQA, ARC, and CSQA only contain the question. All other results use modified prompts with choices provided as part of the question.

| Model | Training | Eval. Mode | PIQA | | ARC | | CSQA | | SIQA | |
|---|---|---|---|---|---|---|---|---|---|---|
| | | | Rand. | Adv. | Rand. | Adv. | Rand. | Adv.† | Rand. | Adv. |
| Random | | - | 50.00 | 50.00 | 25.00 | 25.00 | 20.00 | 20.00 | 33.33 | 33.33 |
| Causal Mask+PE* | Pretrained | Single run | 76.77 | | 37.80 | | 51.76 | | 37.26 | |
| Causal Mask+PE* | Finetuned | Single run | 79.82 | | 45.39 | | 68.80 | | 75.95 | |
| Causal Mask+PE$^{Ultra}$ | Finetuned | No Options | 79.87 | | 43.26 | | 69.37 | | 56.55 | |
| Causal Mask+PE | Pretrained | Single run | 57.45 | 30.96 | 36.03 | 7.68 | 34.92 | 16.46 | 39.29 | 12.74 |
| Causal Mask+PE | Pretrained | Majority Vote | 56.04 | 56.04 | 40.10 | 40.10 | 35.22 | 35.22 | 40.23 | 40.23 |
| Causal Mask+PE | Finetuned | Single run | **84.11** | 76.77 | 55.20 | 23.72 | 78.31 | 69.62 | 74.80 | 63.00 |
| Causal Mask+PE | Finetuned | Majority Vote | 84.06 | **84.06** | 58.87 | 58.87 | 78.38 | 78.38 | **76.05** | **76.05** |
| Causal Mask+PE$^{Ultra}$ | Finetuned | Single run | 83.98 | 77.31 | 56.32 | 26.88 | 77.89 | 68.47 | 74.33 | 63.97 |
| Causal Mask+PE$^{Ultra}$ | Finetuned | Majority Vote | 83.57 | 83.57 | **59.56** | **59.56** | **78.46** | **78.46** | 75.23 | 75.23 |

*Results with original prompts    $^{Ultra}$Additional pretraining    †Only first 24 (of 120) permutations tested

# 5   Experiments and results

## 5.1   Baselines and order sensitivity

We first run baseline models and measure the gap between random-order and adversarial-order accuracies. We use Gemma 2B as the base LLM in the first experiments.

Table 1 shows the results for all Gemma 2B baseline models. Finetuning Gemma 2B on the datasets (Causal Mask+PE Finetuned) gives competitive results in the random evaluation mode. However, adversarial permutations lead to large accuracy drops, particularly for ARC-Challenge (55.20% to 23.72%). In contrast, the majority vote baselines do not have any drops in accuracy between the two evaluation modes, confirming that they are permutation-invariant. Moreover, they also produce the best random-order results. However, this comes at a high cost, as the models have to be run $k!$ times for each input. We use Causal Mask+PE$^{Ultra}$ Finetuned as the baseline for further experiments.

## 5.2   Set-LLM step-by-step

As described in Section 3, there are four steps in turning a base LLM into a Set-LLM. To gain a better understanding of the individual steps, we run experiments with the intermediate models: *Causal Mask+NoPE*, *Prefix Mask+NoPE*, *Prefix Mask+SetPE*, and *SetMask+SetPE* (Set-LLM). Finally, we include *Prefix Mask+PE* in our experiments, which is an encoder-decoder version of the base LLM. Note that in addition to Set-LLM, Prefix Mask+NoPE and Prefix Mask+SetPE are also set-permutation-invariant models and therefore perform exactly the same in the two modes.

Table 2 shows the results for all intermediate models. The set-permutation-invariant models do not have any drops in accuracy in the adversarial setting, confirming our design choices and theoretical results. Moreover, Set-LLM outperforms the strongest baseline on all four benchmarks in both modes, indicating that the permutation-invariance guarantees do not come at a cost to accuracy. This is all the more impressive considering these results come from a single run, rather than $k!$ runs. Prefix Mask+SetPE is not far off SetMask+SetPE, but SetMask is needed to outperform the majority vote baseline and give the best results. The problem illustrated in Figure 4 might help to explain this gap. Additional analyses of the majority vote versus Set-LLM output can be found in Section E.8.

## 5.3   Different base LLMs

In addition to Gemma 2B, we evaluate Set-LLM using Gemma 7B, Llama 3.2 1B, Llama 3.2 3B, and Llama 3.1 8B as base models. Table 3 shows that all base models suffer from order sensitivity with drops between the two evaluation modes ranging from 3.4% to 31.7%. In contrast, there are no drops with Set-LLM, and Set-LLM outperforms the base model in 20/20 cases with adversarial ordering and in 18/20 cases with random ordering. Moreover, with a single run, it outperforms majority vote

Table 2: Set-LLM and all intermediate models going from the base model (Causal Mask+PE) to Set-LLM (SetMask+SetPE). The 4 adaptation steps are described in Section 3. The base model is Gemma 2B. All models are finetuned separately for each benchmark. All scores are accuracies (%).

| Steps | Model | Eval. Mode | # Runs | PIQA Rand. | Adv. | ARC Rand. | Adv. | CSQA Rand. | Adv.[†] | SIQA Rand. | Adv. |
|---|---|---|---|---|---|---|---|---|---|---|---|
| - | Causal Mask+PE[Ultra] | Majority Vote | $k!$ | 83.57 | 83.57 | 59.56 | 59.56 | 78.46 | 78.46 | 75.23 | 75.23 |
| - | Causal Mask+PE | Single run | 1 | 84.11 | 76.77 | 55.20 | 23.72 | 78.31 | 69.62 | 74.80 | 63.00 |
| 1 | Causal Mask+NoPE | Single run | 1 | 74.37 | 63.55 | 35.70 | 14.76 | 68.49 | 57.33 | 63.21 | 48.57 |
| 1,2 | Prefix Mask+NoPE | Single run | 1 | 74.54 | 74.54 | 32.08 | 32.08 | 49.14 | 49.14 | 51.02 | 51.02 |
| 2 | Prefix Mask+PE | Single run | 1 | 82.78 | 76.50 | 57.62 | 27.47 | 78.98 | 71.01 | 74.36 | 65.66 |
| 1-3 | Prefix Mask+SetPE | Single run | 1 | 81.23 | 81.23 | 51.28 | 51.28 | 77.31 | 77.31 | 71.24 | 71.24 |
| 1-4 | SetMask+SetPE | Single run | 1 | 84.33 | 84.33 | 57.76 | 57.76 | 79.93 | 79.93 | 75.38 | 75.38 |
| 1-4 | SetMask+SetPE[Ultra] | Single run | 1 | **85.80** | **85.80** | **65.02** | **65.02** | **80.18** | **80.18** | **76.15** | **76.15** |

[Ultra]Additional pretraining    [†]Only first 24 (of 120) permutations tested    $k$ = number of (multiple) choices

Table 3: Set-LLM performance with different base LLMs. All models were pretrained on UltraFeedback [7] and then finetuned separately for each benchmark. All scores are accuracies (%).

| LLM | Model | Eval. Mode | PIQA Rand. | Adv. | ARC Rand. | Adv. | CSQA Rand. | Adv.[†] | SIQA Rand. | Adv. |
|---|---|---|---|---|---|---|---|---|---|---|
| Gemma 2B | Causal Mask+PE[Ultra] | Single run | 83.98 | 77.31 | 56.32 | 26.88 | 77.89 | 68.47 | 74.33 | 63.97 |
| | Causal Mask+PE[Ultra] | Majority Vote | 84.17 | 84.17 | 60.15 | 60.15 | 78.71 | 78.71 | 75.38 | 75.38 |
| | SetMask+SetPE[Ultra] | Single run | **85.80** | **85.80** | **65.02** | **65.02** | **80.18** | **80.18** | **76.15** | **76.15** |
| Gemma 7B | Causal Mask+PE[Ultra] | Single run | 92.82 | 89.45 | 83.52 | 64.33 | 85.45 | 79.12 | 80.93 | 74.10 |
| | Causal Mask+PE[Ultra] | Majority Vote | 92.66 | 92.66 | **85.58** | **85.58** | **85.75** | **85.75** | 81.10 | 81.10 |
| | SetMask+SetPE[Ultra] | Single run | **92.98** | **92.98** | 83.45 | 83.45 | 84.93 | 84.93 | **81.12** | **81.12** |
| Llama 3.2 1B | Causal Mask+PE[Ultra] | Single run | 79.57 | 71.33 | 53.61 | 21.93 | 74.50 | 64.21 | 71.84 | 62.79 |
| | Causal Mask+PE[Ultra] | Majority Vote | 79.49 | 79.49 | 57.17 | 57.17 | 75.51 | 75.51 | 71.85 | 71.85 |
| | SetMask+SetPE[Ultra] | Single run | **81.66** | **81.66** | **59.30** | **59.30** | **76.66** | **76.66** | **72.47** | **72.47** |
| Llama 3.2 3B | Causal Mask+PE[Ultra] | Single run | 86.92 | 81.72 | 74.16 | 53.07 | 81.32 | 74.94 | 77.54 | 70.42 |
| | Causal Mask+PE[Ultra] | Majority Vote | 86.83 | 86.83 | **76.37** | **76.37** | 81.57 | 81.57 | 77.99 | 77.99 |
| | SetMask+SetPE[Ultra] | Single run | **88.41** | **88.41** | 75.85 | 75.85 | **83.29** | **83.29** | **80.30** | **80.30** |
| Llama 3.1 8B | Causal Mask+PE[Ultra] | Single run | 90.81 | 86.29 | 83.04 | 64.51 | 83.96 | 77.89 | 80.77 | 73.90 |
| | Causal Mask+PE[Ultra] | Majority Vote | 90.75 | 90.75 | **85.32** | **85.32** | 84.11 | 84.11 | 81.12 | 81.12 |
| | SetMask+SetPE[Ultra] | Single run | **91.62** | **91.62** | 84.13 | 84.13 | **85.34** | **85.34** | 81.47 | **81.47** |

[Ultra]Additional pretraining    [†]Only first 24 (of 120) permutations tested

in 16/20 cases, without the exponential runtime overhead. Set-LLM outperforms the baselines across all model architectures and sizes we tested. Additional results can be found in Sections E.3 and E.4.

## 5.4 Out-of-distribution performance

When using Set-LLM as an LLM evaluator, it is particularly important that the adaptations do not hurt the performance on out-of-distribution data after finetuning on a small dataset. We measure the out-of-distribution multiple-choice performance of the models by finetuning a model on each benchmark independently and then evaluating on the three remaining benchmarks. We do this for both the base model and Set-LLM and compare their performance with the pretrained base model. Results are provided in Table 18 in the Appendix. Set-LLM is the best-performing model in 10/12 cases in both evaluation modes. We also measure the impact of the Set-LLM adaptations on standard (non-set) language task performance. Please refer to Section E.6 for these experiments.

Table 4: Multi-document summarization with different base LLMs. All models were pretrained on UltraFeed-back [7] and then finetuned on MultiNews [9]. All Rouge metrics are F1 scores.

| LLM | Model | Compression | Rouge-1 | Rouge-2 | Rouge-1* | Rouge-2* |
|---|---|---|---|---|---|---|
| Target Summaries | | 6.09 | 0.27 | 0.14 | - | - |
| Gemma 2B | Causal Mask+PE$^{\text{Ultra}}$ | 15.84 | 0.17 | 0.12 | 0.33 | 0.11 |
| | SetMask+SetPE$^{\text{Ultra}}$ | 7.23 | **0.27** | **0.20** | **0.47** | **0.19** |
| Llama 3.2 1B | Causal Mask+PE$^{\text{Ultra}}$ | 20.76 | 0.13 | 0.08 | 0.33 | 0.10 |
| | SetMask+SetPE$^{\text{Ultra}}$ | 7.31 | **0.27** | **0.18** | **0.48** | **0.18** |
| Llama 3.2 3B | Causal Mask+PE$^{\text{Ultra}}$ | 9.77 | 0.23 | 0.15 | 0.43 | 0.15 |
| | SetMask+SetPE$^{\text{Ultra}}$ | 7.23 | **0.27** | **0.18** | **0.49** | **0.20** |

*Rouge F1 scores between the target summaries and the model summaries.

## 6 Potential use cases

We include additional proof-of-concept experiments to demonstrate the benefit of Set-LLM in three different LLM applications: multi-document summarization, multi-document question answering, and LLM-as-a-judge (Section E.10). We see promising results in all three use cases.

### 6.1 Multi-document summarization

In multi-document summarization the LLM is presented with a set of documents and tasked with producing a collective summary. We perform our evaluation on the MultiNews dataset [9]. Details about the experimental setup can be found in Section D.5.

We report standard summarization metrics, namely n-gram Rouge F1 scores between the model summaries and the input documents and compression rate – the length of the original text divided by the length of the summary. We also include Rouge F1 scores between the target summaries and the model summaries – Rouge-1* and Rouge-2*. The results are summarized in Table 4. Set-LLM outperforms the finetuned baseline with all three base models in all metrics. Note that Set-LLM produces longer summaries, which leads to lower compression rates.

### 6.2 Multi-document question answering

Motivated by [27], we evaluate Set-LLM in answering multi-document questions. The task consists of questions with 20 supporting documents, where only one document contains the information required for the answer. The authors show that many LLMs (including commercial models) produce a U-shaped performance curve with respect to the location of the answer. If the relevant document is listed first or last, then the LLM correctly answers the question with a higher probability. In contrast, Set-LLM is guaranteed to have a flat curve, since reordering the documents does not affect the output. We test five different locations (1st, 5th, 10th, 15th, and last) for the placement of the answer document. We provide results for two LLama-based models and compare these with GPT-3.5-Turbo scores from [27]. Details about the experimental setup can be found in Section D.6.

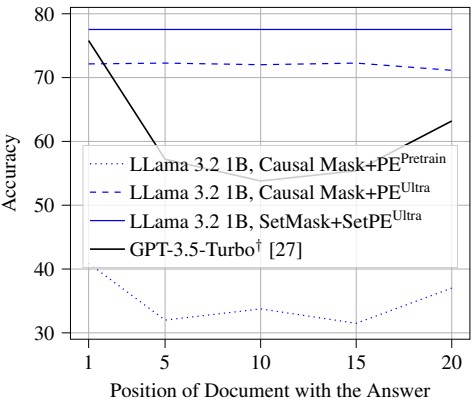

Figure 5: Varying the position of the relevant document (containing the answer) within the pretrained model's input results in a U-shaped performance curve. However, Set-LLM (or finetuning) produces flat performance curves with a higher accuracy. †GPT-3.5-Turbo was not finetuned on the dataset.

As expected, the results in Figure 5 demonstrate that Set-LLM is not sensitive to the location of the relevant document. Furthermore, Set-LLM outperforms the baseline model, even after finetuning. The pretrained base model produces a U-shaped curve in line with previous work [27], but the

finetuned version produces a mostly flat curve. For larger models, the advantage of Set-LLM over the finetuned base model seems to diminish. For results with Llama 3.2 3B, refer to Section E.9.

# 7 Related work

To the best of our knowledge, this is the first paper to specifically introduce a permutation-invariant architecture for decoder-only language models. Concurrently to our work, Kinder et al. [19] propose Set-Encoding to address positional bias in LLMs. Similar to our work they propose to modify the positional encoding and attention mask for set-valued input.

**Permuting multiple-choice questions.** This work was motivated by the observations that you can "fool your (vision and) language model with embarrassingly simple permutations" [57]. The authors quantify the effect of adversarial permutations on (V)LLMs and multiple-choice benchmarks. They also analyze the effectiveness of majority vote, which we include as a baseline in our experiments. Prior to this, Liu et al. [29] suggested rotating the choices of multiple-choice questions to evaluate the robustness of (Multimodal) Language Models. Taking a majority vote over the rotations could be used as an alternative to majority vote, thereby only carrying a linear-factor rather than an exponential-factor runtime overhead, but it is not guaranteed to give the same answer for all permutations.

**LLM evaluators.** LLM evaluators (or judges) were introduced recently [53, 55], but have already been integrated in many LLM pipelines, including for evaluation, retrieval and reasoning [2, 10, 12, 26]. The order bias of LLM evaluators was already observed by Zheng et al. [53], and methods involving the aggregation of multiple runs and specialized prompting have been proposed to mitigate this problem [18, 29, 41, 56, 57]. However, this is the first work to eliminate order bias directly from the model architecture without impacting accuracy or runtime.

**Graphs and large language models.** A set can be seen as an empty (or fully-connected) graph, and indeed one of the key features of graphs is also permutation-invariance. As a result, several works that combine graphs and LLMs are also relevant here [14, 48, 49]. Most closely related to this work are the papers of Liu et al. [28] and Plenz and Frank [33], who adapt an encoder-decoder and an encoder-only language model, respectively, to take mixed graph-text data as input. Both approaches alter the attention mechanism and the positional encoding to incorporate graph connectivity information into the embeddings.

**Invariant neural networks.** While invariant networks have existed for some time, Bronstein et al. [4] recently unified them under a common geometric framework. For example, CNNs [23, 52] are shift-invariant architectures for images, Deep Sets [50] was the first permutation-invariant architecture for sets, and GNNs [20, 44, 47] are permutation-invariant architectures for graphs. In this work, we combine the permutation invariance of these architectures with the power of large language models.

# 8 Conclusion

To the best of our knowledge, this paper introduces the first permutation-invariant decoder-only LLM, Set-LLM. We formally prove that Set-LLM is permutation-invariant and show how robust it is to permutations in practice. The models are based on pretrained LLMs, can be finetuned efficiently with low-rank adapters, and incur no additional runtime costs. We test Set-LLM on four multiple-choice datasets, where it outperforms all baselines by significant margins in either accuracy or runtime. Moreover, we show promising results using Set-LLM in two important applications: multi-document summarization and multi-document question answering.

With the growing importance of multi-document applications, Set-LLM has the potential for wide impact. We provide promising initial experiments for natural use cases, and we hope that future work can explore these directions in detail.

**Limitations.** While we propose a general purpose set-permutation-invariant LLM, the approach requires explicit knowledge of which parts of the input are set-valued, which may not be readily available in some applications. There is no obvious way to combine Set Mask with sparse attention mechanisms, e.g. sliding window attention. This would be an interesting direction for future work.

## Acknowledgments

The authors would like to thank the Vector Stiftung for their financial support in the framework of the MINT Innovations program.

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

# A  Impact

We do not foresee any direct negative impacts from this work. On the contrary, we believe this work, as with any other work on model robustness, can contribute positively to LLM applications, especially in high-risk scenarios. Moreover, this work can be used to develop more robust evaluation approaches, which can help move the wider field forward.

# B  SetPE algorithm

---

**Algorithm 1** Python-like pseudocode for calculating Set Position Encoding (SetPE) positions.

---

```
# input - mixed list of token ids and sets of token id lists
# vocab - token vocabulary

pos = [ ] # list of SetPE positions
ind = 0 # running position index
for q in input:  # iterate through elements of the input
  if q in vocab:  # if element q is a token
    pos.append(ind)
    ind += 1
  else:  # if element q is a set of token lists
    for s in q:  # iterate through set of token lists
      # append consecutive positions for each token in s:
      pos = pos + list(range(ind, ind+len(s)))
    total_tokens_q = sum([len(s) for s in q])
    ind += total_tokens_q
return pos
```

---

# C  Proofs

**Theorem 1** (Set Permutation Equivariance). *Let $\pi$ be a permutation corresponding to permuting elements of sets in a mixed set-text input, and let $P$ be the corresponding permutation matrix. If $X^{(t+1)} = A^{(t)} X^{(t)} W_V$ is the output of an attention layer with SetPE, SetMask, and absolute positional encoding, then $\tilde{X}^{(t+1)} = P X^{(t+1)}$. In other words, attention with SetPE and SetMask is equivariant to set permutations of mixed set-text input.*

*Proof of Theorem 1.*

**Claim 2.1.** $\tilde{X} = PX$

First note, that by the definition of SetPE, SetPE positions will be the same before and after the set permutation is applied to the input. This is because all token sequences remain in the same set after a set permutation, and all token sequences within a set are numbered consecutively from the same starting index. If $\tilde{}$ denotes variables after permuting the input, then $\tilde{x}_{\pi(i)} = \psi(t_i, \text{pos}[i]) = x_i$. Tokens that are not part of a set are not moved by a set permutation, $\pi(i) = i$ for these tokens and again $\tilde{x}_{\pi(i)} = \psi(t_i, \text{pos}[i]) = x_i$. Therefore, $\tilde{X} = PX$. □

**Claim 2.2.** $\tilde{Z} = PZP^T$

By Claim 2.1, we have

$$
\begin{aligned}
\tilde{Z} &= attn(\tilde{X}, \tilde{X}, W_Q, W_K, d_K) \\
&= attn(PX, PX, W_Q, W_K, d_K) \\
&= PXW_Q(PXW_K)^T / \sqrt{d_K} \\
&= PXW_Q W_K^T X^T P^T / \sqrt{d_K} \\
&= P\left(XW_Q W_K^T X^T / \sqrt{d_K}\right) P^T \quad = PZP^T.
\end{aligned}
$$

$\square$

**Claim 2.3.** $\tilde{M} = PMP^T$

Though this is not the case for general permutations of the tokens, we show that this is the case when $\pi$ is restricted to permuting elements within the sets $q_i$.

The claim is equivalent to showing that $\tilde{M}_{\pi(i)\pi(j)} = M_{ij}$ for all $i, j$. This can be seen from the construction of SetMask and the definition of $\pi$. But to be more precise, we split the claim into multiple cases:

Recall that $s(t_i)$ denotes the token sequence containing the token $t_i$ and $q(t_i)$ denotes the set containing $s(t_i)$.

**Case 1:** $q(t_i) \neq q(t_j)$. $\quad q(t_i) \neq q(t_j) \iff q(t_{\pi(i)}) \neq q(t_{\pi(j)})$ since $\pi$ does not move tokens between sets. Then by definition of SetMask, $\tilde{M}_{\pi(i)\pi(j)} = 1 = M_{ij}$.

**Case 2:** $q(t_i) = q(t_j), s(t_i) = s(t_j)$. $\quad$ Since $\pi$ permutes whole sequences within a set together, all tokens within a sequence remain in the same sequence after permutation. Therefore, $\tilde{M}_{\pi(i)\pi(j)} = 1 = M_{ij}$.

**Case 3:** $q(t_i) = q(t_j), s(t_i) \neq s(t_j)$. $\quad$ Again, since $\pi$ permutes whole sequences within a set together, tokens in different sequences of the same set remain in different sequences of the same set after permutation. Therefore, $\tilde{M}_{\pi(i)\pi(j)} = 0 = M_{ij}$. $\hfill\square$

**Claim 2.4.** $\tilde{A} = PAP^T$

Again, this claim is equivalent to showing that $\tilde{A}_{\pi(i)\pi(j)} = A_{ij}$ for all $i, j$. Putting the previous claims together, we have:

**Case 1:** $M_{ij} = 0$. $\quad$ By Claim 2.3, if $M_{ij} = 0$, then $\tilde{M}_{\pi(i)\pi(j)} = 0$ and $A_{ij} = \tilde{A}_{\pi(i)\pi(j)} = 0$.

**Case 2:** $M_{ij} = 1$. $\quad$ By Claim 2.3, if $M_{ij} = 1$, then $\tilde{M}_{\pi(i)\pi(j)} = 1$, and we have

$$\tilde{A}_{\pi(i)\pi(j)} = \frac{exp(\tilde{Z}_{\pi(i)\pi(j)})}{\sum_k \tilde{M}_{\pi(i)\pi(k)}exp(\tilde{Z}_{\pi(i)\pi(k)})} \tag{8}$$

$$\overset{\text{(by Claims 2.2 and 2.3)}}{=} \frac{exp(Z_{ij})}{\sum_k M_{ik}exp(Z_{ik})} = A_{ij} \tag{9}$$

$\hfill\square$

Finally, putting all the claims together, we have:

$$\tilde{X}^{(t+1)} = \tilde{A}^{(t)}\tilde{X}^{(t)}W_V \tag{10}$$

$$\overset{\text{(by Claims 2.1 and 2.4)}}{=} PA^{(t)}P^T PX^{(t)}W_V \tag{11}$$

$$= PA^{(t)}X^{(t)}W_V \tag{12}$$

$$= PX^{(t+1)}, \tag{13}$$

completing the proof that the attention layer is equivariant. $\hfill\square$

**Theorem 2.** *Attention with SetPE and SetMask reduces to attention with PE and prefix masking when the input is a single sequence of tokens (i.e., when the input does not contain sets).*

*Proof of Theorem 2.* If the input is a single sequence of tokens, we have $q = [q_0]$, $q_0 = [s_0]$, and $s_0 = [\tau_0, \tau_0, \ldots, \tau_{n_{0,0}}]$. In this case, Algorithm 1 assigns consecutive positions for each token in the input, that is, $[0, 1, 2, \ldots, n_{0,0}]$. This is identical to absolute positions (PE).

Similarly, if there are no sets in the input, then SetMask reduces to a fully connected attention mask on the input, i.e., to the prefix mask. $\hfill\square$

# D  Additional experimental details

## D.1  Instruction finetuning

Finally, we introduce *Instruction Finetuning* as the proposed method relies on this training approach. Many LLMs use some form of instruction tuning during the (pre-)training process [32, 51]. It involves a dataset $\mathcal{D}$ of instruction-answer (or prompt-response) pairs $\mathcal{D} = \{(q, a)\}_{i=1}^{N}$. The training objective is to maximize the probability of autoregressively generating $a$ given $q$, i.e., to maximize

$$p(a \mid q) = \prod_{i=1,\ldots,|a|} p(a_i \mid q, a_1, \ldots, a_{i-1}). \tag{14}$$

We use instruction tuning for training all our models to align to the new position encoding and attention masking setups. Since all probabilities in the product are conditional on $q$, $q$ can be encoded without a causal mask, i.e., tokens in $q$ could attend to earlier tokens in $q$. This makes instruction tuning ideal for our scenario.

## D.2  Dataset details

We get all the datasets from HuggingFace Datasets [24]. Table 5 provides metadata and Table 6 provides licensing details for each dataset.

Table 5: HuggingFace Datasets path, number of train/evaluation samples, number of choices per question, and number of answer choice permutations for each dataset in this paper.

| Dataset | Path | # Train. Samples | # Eval. Samples | $k$ | $k!$ |
|---|---|---|---|---|---|
| UltraFeedback [7, 21] | openbmb/UltraFeedback | | - | - | - |
| PIQA [3] | piqa | 16113 | 3084 | 2 | 2 |
| ARC [5] | allenai/ai2_arc | 1119 | 1172 | 4 | 24 |
| CSQA [40] | tau/commonsense_qa | 9741 | 1140 | 5 | 120 |
| SIQA [36] | social_i_qa | 33410 | 1954 | 3 | 6 |
| MultiNews [9] | alexfabbri/multi_news | 44972 | 5620 | 2-10 | - |
| Natural QA [22, 27] | - | 7965 | 2655 | 20 | 2.4e+18 |

Table 6: Dataset Licenses.

| Dataset | License |
|---|---|
| UltraFeedback [7, 21] | MIT license |
| PIQA [3] | unknown |
| ARC [5] | CC-BY-SA 4.0 |
| CSQA [40] | MIT license |
| SIQA [36] | unknown |
| MultiNews [9] | other |
| Natural QA [22, 27] | CC-BY-SA 3.0 |

## D.3 Prompt templates

**Original templates.** We provide all the original templates used to create prompt-response pairs for each dataset, for both training and evaluation.

Table 7: Original prompt templates for each dataset.

| Dataset | Original Prompt | Response |
|---------|-----------------|----------|
| UltraFeedback [7] | Question: {instruction}

Answer: | {answer}<EOS> |
| PIQA [3] | Question: {question}

Answer: | {answer}<EOS> |
| ARC [5] | Question: {question}

Answer: | {answer}<EOS> |
| CSQA [40] | Question: {question}

Answer: | {answer}<EOS> |
| SIQA [36] | Question: Given the context, answer correctly the question.
Context: {context}
Question: {question}

Choices:
(0) {choice0}
(1) {choice1}
(2) {choice2}

Answer: | ({answer_index})<EOS> |

**Modified templates.** We provide all the modified templates used to create prompt-response pairs for each dataset, for both training and evaluation.

Table 9: Modified prompt templates for each dataset.

| Dataset | Modified Prompt | Response |
|---|---|---|
| UltraFeedback [7] | Question: {instruction}

Answer: | {answer}<EOS> |
| PIQA [3] | Question: {question}

Choices:
{choice0}
{choice1}

Answer: | {answer}<EOS> |
| ARC [5] | Question: {question}

Choices:
{choice0}
{choice1}
{choice2}
{choice3}

Answer: | {answer}<EOS> |
| CSQA [40] | Question: {question}

Choices:
{choice0}
{choice1}
{choice2}
{choice3}
{choice4}

Answer: | {answer}<EOS> |
| SIQA [36] | Question: Given the context, answer correctly the question.
Context: {context}
Question: {question}

Choices:
{choice0}
{choice1}
{choice2}

Answer: | {answer}<EOS> |

### D.4 Hyperparameter settings

Table 11: Hyperparameters shared across models and datasets.

| Hyperparameter | Value |
|---|---|
| GPUs | 1 |
| Optimizer | AdamW |
| LR Scheduler | Linear |
| Weight Decay | 0.0 |
| Batch Size | 10 |
| Accumulation Steps | 10 |
| Warmup Steps | 300 (or 10% of update steps) |
| Update Steps | 3000 |
| Random Seed | 42 |

We tune the learning rate for each model and dataset using a logarithmic scale: [1e-4, 3e-4, 1e-3, 3e-3]. The final (best) learning rates are presented in Table 12.

Table 12: Learning rates for all final baseline and Set-LLM models on all datasets.

| LLM | Model | Ultra. | PIQA | ARC | CSQA | SIQA | MultiNews | Natural QA |
|---|---|---|---|---|---|---|---|---|
| Gemma 2B | Causal Mask+PE$^{\text{Ultra}}$ | 3e-4 | 1e-3 | 1e-3 | 1e-3 | 1e-3 | 1e-3 | 1e-3 |
| | SetMask+SetPE$^{\text{Ultra}}$ | 3e-4 | 1e-3 | 1e-3 | 1e-3 | 1e-3 | 1e-3 | 1e-3 |
| Gemma 7B | Causal Mask+PE$^{\text{Ultra}}$ | 3e-4 | 3e-4 | 3e-4 | 3e-4 | 3e-4 | 3e-4 | 3e-4 |
| | SetMask+SetPE$^{\text{Ultra}}$ | 3e-4 | 3e-4 | 3e-4 | 3e-4 | 3e-4 | 3e-4 | 3e-4 |
| Llama 3.2 1B | Causal Mask+PE$^{\text{Ultra}}$ | 1e-3 | 1e-3 | 1e-3 | 1e-3 | 1e-3 | 1e-3 | 1e-3 |
| | SetMask+SetPE$^{\text{Ultra}}$ | 1e-3 | 1e-3 | 1e-3 | 1e-3 | 1e-3 | 1e-3 | 1e-3 |
| Llama 3.2 3B | Causal Mask+PE$^{\text{Ultra}}$ | 3e-4 | 1e-3 | 1e-3 | 1e-3 | 1e-3 | 1e-3 | 1e-3 |
| | SetMask+SetPE$^{\text{Ultra}}$ | 3e-4 | 1e-3 | 1e-3 | 1e-3 | 1e-3 | 1e-3 | 1e-3 |
| Llama 3.1 8B | Causal Mask+PE$^{\text{Ultra}}$ | 3e-4 | 3e-4 | 3e-4 | 3e-4 | 3e-4 | 3e-4 | 3e-4 |
| | SetMask+SetPE$^{\text{Ultra}}$ | 1e-3 | 3e-4 | 3e-4 | 3e-4 | 3e-4 | 3e-4 | 3e-4 |

Table 13: We use LoRA [15] to train all our models. We use the HuggingFace PEFT library [30] with default hyperparameter values, unless listed.

| Hyperparameter | Value |
|---|---|
| Rank | 8 |
| Alpha | 1 |
| Target Modules | All linear layers of MLP and Self-Attention |

### D.5 Multi-document summarization

We perform our evaluation on the MultiNews dataset [9]. We filter out inputs of length greater than 20000 characters to satisfy our memory constraints. We use the standard train-validation-test split, but subsample 1000 test samples. We compare the original model architectures with Set-LLM, performing a single model run in both cases (no adversarial setting). We use the best-performing hyperparameters from the multiple-choice experiments for finetuning.

We report standard summarization metrics, namely Rouge F1 scores between the model summaries and the input documents and compression rate (the length of the original text divided by the length of the summary). We also include Rouge F1 scores between the target summaries and the model summaries (R1*, R2*, R3*).

### D.6 Multi-document question answering

We evaluate Set-LLM in multi-document question answering. We use the dataset from [27] based on the natural questions benchmark [22]. The task consists of questions with 20 supporting documents, where only one document contains the information required for the answer. We use the original train-validation-test split from [22]: 60% training, 10% validation, and 30% test data. We replicate the training data five times and shuffle the documents at random for each input. This is done to avoid overfitting the model to a specific formulation of each training question. We use the best-performing hyperparameters from the multiple-choice experiments for finetuning. We have five different test sets with the answer document placed in five different locations (1st, 5th, 10th, 15th, and last). We provide results for two LLama-based models. All scores denote the percentage accuracy.

## E Additional experimental results

### E.1 Additional pretraining

To help the models adapt to the architectural changes, we experiment with additional pretraining. We use a high-quality subset (approximately 10k examples) of the cleaned UltraFeedback dataset [7], attained by following the data preprocessing steps in [21]. This additional pretraining aims to help the new, adapted model architectures better adapt to the architectural changes. We therefore hypothesize that adapted models will benefit more from this data than the unaltered baseline models.

We present complete results with and without additional pretraining in Table 14. Consistent with our hypothesis, the additional data significantly improves the performance of Prefix Mask+PE, Prefix Mask+SetPE, and SetMask+SetPE, especially on ARC-Challenge, but does not improve the performance of the finetuned Causal Mask+PE. However, the pretrained Causal Mask+PE results improve the most, suggesting that the pretrained base model is not particularly well-suited to the benchmark task setups.

Table 14: Results of (pre-)finetuning Gemma 2B on UltraFeedback with the respective attention mask and positional encoding. The $*$ indicates results using the original dataset prompts, which for the PIQA, ARC, and CSQA benchmark only contain the question. All other results use modified prompts, where the choices are provided with the question. $\dagger$The CSQA dataset has exactly 5 choices for each question, but we run the adversarial search for only 24 permutations. All scores are accuracies (%).

| Model | Training | PIQA | | ARC | | CSQA | | SIQA | |
|---|---|---|---|---|---|---|---|---|---|
| | | Std. | Adv. | Std. | Adv. | Std. | Adv.$\dagger$ | Std. | Adv. |
| Causal Mask+PE$^*$ | Pretrained | 76.77 | | 37.80 | | 51.76 | | 37.26 | |
| Causal Mask+PE$^*$ | Finetuned | 79.82 | | 45.39 | | 68.80 | | 75.95 | |
| Causal Mask+PE | Pretrained | 57.45 | 30.96 | 36.03 | 7.68 | 34.92 | 16.46 | 39.29 | 12.74 |
| Causal Mask+PE$^{Ultra}$ | Pretrained | 68.31 | 50.49 | 43.18 | 14.93 | 45.21 | 27.35 | 46.72 | 16.84 |
| Causal Mask+PE | Finetuned | 84.11 | 76.77 | 55.20 | 23.72 | 78.31 | 69.62 | 74.80 | 63.00 |
| Causal Mask+PE$^{Ultra}$ | Finetuned | 83.98 | 77.31 | 56.32 | 26.88 | 77.89 | 68.47 | 74.33 | 63.97 |
| Prefix Mask+PE | Finetuned | 82.78 | 76.50 | 57.62 | 27.47 | 78.98 | 71.01 | 74.36 | 65.66 |
| Prefix Mask+PE$^{Ultra}$ | Finetuned | 84.93 | 79.11 | 61.26 | 34.47 | 79.28 | 70.19 | 75.18 | 67.45 |
| Prefix Mask+SetPE | Finetuned | 81.23 | 81.23 | 51.28 | 51.28 | 77.31 | 77.31 | 71.24 | 71.24 |
| Prefix Mask+SetPE$^{Ultra}$ | Finetuned | 81.88 | 81.88 | 56.48 | 56.48 | 77.97 | 77.97 | 73.54 | 73.54 |
| SetMask+SetPE | Finetuned | 84.33 | 84.33 | 57.76 | 57.76 | 79.93 | 79.93 | 75.38 | 75.38 |
| SetMask+SetPE$^{Ultra}$ | Finetuned | **85.80** | **85.80** | **65.02** | **65.02** | **80.18** | **80.18** | **76.15** | **76.15** |

$^*$Results with original prompts    $^{Ultra}$Additional pretraining    $\dagger$Only first 24 (of 120) permutations tested

## E.2 Majority vote

Additional results with majority vote are presented in Table 15.

Table 15: Majority Vote results with Gemma 2B as the base model. The $*$ indicates results using the original dataset prompts, which for the PIQA, ARC, and CSQA benchmark only contain the question. All other results use modified prompts, where the choices are provided with the question. $\dagger$ The CSQA dataset has exactly $5$ choices for each question, but we run the adversarial search for only $24$ permutations. All scores are accuracies ($\%$).

| Model | Training | Eval. Mode | PIQA | | ARC | | CSQA | | SIQA | |
|---|---|---|---|---|---|---|---|---|---|---|
| | | | Std. | Adv. | Std. | Adv. | Std. | Adv.$\dagger$ | Std. | Adv. |
| Causal Mask+PE$^*$ | Pretrained | Single run | 76.77 | | 37.80 | | 51.76 | | 37.26 | |
| Causal Mask+PE$^*$ | Finetuned | Single run | 79.82 | | 45.39 | | 68.80 | | 75.95 | |
| Causal Mask+PE$^{\text{Ultra}}$ | Pretrained | Single run | 68.31 | 50.49 | 43.18 | 14.93 | 45.21 | 27.35 | 46.72 | 16.84 |
| Causal Mask+PE$^{\text{Ultra}}$ | Pretrained | Majority Vote | 68.12 | 68.12 | 46.16 | 46.16 | 45.95 | 45.95 | 48.62 | 48.62 |
| Causal Mask+PE$^{\text{Ultra}}$ | Finetuned | Single run | 83.98 | 77.31 | 56.32 | 26.88 | 77.89 | 68.47 | 74.33 | 63.97 |
| Causal Mask+PE$^{\text{Ultra}}$ | Finetuned | Majority Vote | 83.57 | 83.57 | 59.56 | 59.56 | 78.46 | 78.46 | 75.23 | 75.23 |
| Causal Mask+NoPE$^{\text{Ultra}}$ | Finetuned | Single run | 75.03 | 64.53 | 37.01 | 14.68 | 68.87 | 57.41 | 65.57 | 51.79 |
| Causal Mask+NoPE$^{\text{Ultra}}$ | Finetuned | Majority Vote | 75.35 | 75.35 | 38.82 | 38.82 | 69.45 | 69.45 | 66.63 | 66.63 |
| Prefix Mask+PE$^{\text{Ultra}}$ | Finetuned | Single run | 84.93 | 79.11 | 61.26 | 34.47 | 79.28 | 70.19 | 75.18 | 67.45 |
| Prefix Mask+PE$^{\text{Ultra}}$ | Finetuned | Majority Vote | 85.36 | 85.36 | 64.08 | 64.08 | 79.77 | 79.77 | 75.84 | 75.84 |
| SetMask+SetPE$^{\text{Ultra}}$ | Finetuned | Single run | **85.80** | **85.80** | **65.02** | **65.02** | **80.18** | **80.18** | **76.15** | **76.15** |

$^*$Results with original prompts    $^{\text{Ultra}}$Additional pretraining    $\dagger$Only first 24 (of 120) permutations tested

## E.3 Different base LLMs

Additional results with different base LLMs are presented in Table 16. Table 17 provides the flip probabilities for the Causal Mask+PE[Ultra] and SetMask+SetPE[Ultra] models. The flip probability is a measure of how inconsistent the model predictions are between different permutations. Please, see Section E.4 for more details.

Table 16: Performance with different base LLMs. All results (except Pretrained*) are (pre-)finetuned on the ultrafeedback dataset. †The CSQA dataset has exactly 5 choices for each question, but we run the adversarial search for only 24 permutations. All scores are accuracies (%).

| LLM | Model | Training | PIQA Rand. | PIQA Adv. | ARC Rand. | ARC Adv. | CSQA Rand. | CSQA Adv.† | SIQA Rand. | SIQA Adv. |
|---|---|---|---|---|---|---|---|---|---|---|
| Gemma 2B | Causal Mask+PE* | Pretrained | 76.77 | | 37.80 | | 51.76 | | 37.26 | |
| | Causal Mask+PE[Ultra] | Pretrained | 68.31 | 50.49 | 43.18 | 14.93 | 45.21 | 27.35 | 46.72 | 16.84 |
| | Causal Mask+PE[Ultra] | Finetuned | 83.98 | 77.31 | 56.32 | 26.88 | 77.89 | 68.47 | 74.33 | 63.97 |
| | + Majority Vote | Finetuned | 84.17 | 84.17 | 60.15 | 60.15 | 78.71 | 78.71 | 75.38 | 75.38 |
| | + No Options | Finetuned | 79.87 | 79.87 | 43.26 | 43.26 | 69.37 | 69.37 | 56.55 | 56.55 |
| | SetMask+SetPE[Ultra] | Finetuned | **85.80** | **85.80** | **65.02** | **65.02** | **80.18** | **80.18** | **76.15** | **76.15** |
| Gemma 7B | Causal Mask+PE* | Pretrained | 80.41 | | 43.77 | | 62.16 | | 65.71 | |
| | Causal Mask+PE[Ultra] | Pretrained | 86.21 | 78.67 | 79.68 | 56.23 | 69.96 | 47.17 | 70.34 | 49.69 |
| | Causal Mask+PE[Ultra] | Finetuned | 92.82 | 89.45 | 83.52 | 64.33 | 85.45 | 79.12 | 80.93 | 74.10 |
| | + Majority Vote | Finetuned | 92.66 | 92.66 | **85.58** | **85.58** | 85.75 | 85.75 | 81.10 | 81.10 |
| | + No Options | Finetuned | 83.51 | 83.51 | 52.47 | 52.47 | 73.79 | 73.79 | 59.47 | 59.47 |
| | SetMask+SetPE[Ultra] | Finetuned | **92.98** | **92.98** | 83.45 | 83.45 | 84.93 | 84.93 | **81.12** | **81.12** |
| Llama 3.2 1B | Causal Mask+PE* | Pretrained | 74.32 | | 35.41 | | 55.77 | | 51.59 | |
| | Causal Mask+PE[Ultra] | Pretrained | 63.03 | 40.48 | 40.12 | 9.13 | 45.29 | 21.21 | 49.27 | 21.39 |
| | Causal Mask+PE[Ultra] | Finetuned | 79.57 | 71.33 | 53.61 | 21.93 | 74.50 | 64.21 | 71.84 | 62.79 |
| | + Majority Vote | Finetuned | 79.49 | 79.49 | 57.17 | 57.17 | 75.51 | 75.51 | 71.85 | 71.85 |
| | + No Options | Finetuned | 77.42 | 77.42 | 39.33 | 39.33 | 65.03 | 65.03 | 53.63 | 53.63 |
| | SetMask+SetPE[Ultra] | Finetuned | **81.66** | **81.66** | **59.30** | **59.30** | **76.66** | **76.66** | **72.47** | **72.47** |
| Llama 3.2 3B | Causal Mask+PE* | Pretrained | 76.33 | | 43.94 | | 61.92 | | 65.97 | |
| | Causal Mask+PE[Ultra] | Pretrained | 76.41 | 64.09 | 68.83 | 39.93 | 66.85 | 44.31 | 66.56 | 47.80 |
| | Causal Mask+PE[Ultra] | Finetuned | 86.92 | 81.72 | 74.16 | 53.07 | 81.32 | 74.94 | 77.54 | 70.42 |
| | + Majority Vote | Finetuned | 86.83 | 86.83 | **76.37** | **76.37** | 81.57 | 81.57 | 77.99 | 77.99 |
| | + No Options | Finetuned | 79.54 | 79.54 | 45.31 | 45.31 | 71.34 | 71.34 | 56.55 | 56.55 |
| | SetMask+SetPE[Ultra] | Finetuned | **88.41** | **88.41** | 75.85 | 75.85 | **83.29** | **83.29** | **80.30** | **80.30** |
| Llama 3.1 8B | Causal Mask+PE* | Pretrained | 80.09 | | 53.41 | | 66.50 | | 69.34 | |
| | Causal Mask+PE[Ultra] | Pretrained | 83.30 | 72.36 | 78.75 | 56.66 | 72.67 | 53.81 | 70.91 | 54.96 |
| | Causal Mask+PE[Ultra] | Finetuned | 90.81 | 86.29 | 83.04 | 64.51 | 83.96 | 77.89 | 80.77 | 73.90 |
| | + Majority Vote | Finetuned | 90.75 | 90.75 | **85.32** | **85.32** | 84.11 | 84.11 | 81.12 | 81.12 |
| | + No Options | Finetuned | 82.43 | 82.43 | 46.33 | 46.33 | 72.97 | 72.97 | 58.60 | 58.60 |
| | SetMask+SetPE[Ultra] | Finetuned | **91.62** | **91.62** | 84.13 | 84.13 | **85.34** | **85.34** | **81.47** | **81.47** |

*Results with original prompts    [Ultra]Additional pretraining    †Only first 24 (of 120) permutations tested

Table 17: We use *flip probabilities* to measure how consistent model predictions are across permutations. They measure how often a model prediction deviates from its *majority vote* for a sample. All models are (pre-)finetuned on the ultrafeedback dataset. [†]The CSQA dataset has exactly 5 choices for each question, but we calculate the majority vote and the flip probabilities using only 24 permutations. All probabilities are given as percentages (%).

| LLM | Model | Training | PIQA | ARC | CSQA[†] | SIQA |
|-----|-------|----------|------|-----|---------|------|
| Gemma 2B | Causal Mask+PE$^{\text{Ultra}}$ | Finetuned | 6.66 | 21.45 | 5.40 | 6.35 |
| | SetMask+SetPE$^{\text{Ultra}}$ | Finetuned | 0.00 | 0.00 | 0.00 | 0.00 |
| Gemma 7B | Causal Mask+PE$^{\text{Ultra}}$ | Finetuned | 3.37 | 7.92 | 2.94 | 4.32 |
| | SetMask+SetPE$^{\text{Ultra}}$ | Finetuned | 0.00 | 0.00 | 0.00 | 0.00 |
| Llama 3.2 1B | Causal Mask+PE$^{\text{Ultra}}$ | Finetuned | 8.24 | 23.37 | 5.72 | 5.70 |
| | SetMask+SetPE$^{\text{Ultra}}$ | Finetuned | 0.00 | 0.00 | 0.00 | 0.00 |
| Llama 3.2 3B | Causal Mask+PE$^{\text{Ultra}}$ | Finetuned | 5.20 | 11.76 | 3.32 | 4.39 |
| | SetMask+SetPE$^{\text{Ultra}}$ | Finetuned | 0.00 | 0.00 | 0.00 | 0.00 |
| Llama 3.1 8B | Causal Mask+PE$^{\text{Ultra}}$ | Finetuned | 4.52 | 7.87 | 3.23 | 4.63 |
| | SetMask+SetPE$^{\text{Ultra}}$ | Finetuned | 0.00 | 0.00 | 0.00 | 0.00 |

### E.4 Flip Probabilities

The **flip probability** is the fraction of times that the predictions differ from the majority vote. Specifically, we run all possible permutations of the input through the LLM and pick the option with the most "votes" as the majority vote. We then count how many of the votes disagree with the majority vote and take the average over all samples. One can see this as a measure of how inconsistent the model predictions are across permutations. Unlike the adversarial setting, where a single permutation of $k!$ can lead to an incorrect outcome, the flip probability is less dependent on the value of $k$.

Table 17 shows the flip probabilities for the finetuned base models and their Set-LLM counterparts from the main table of results, Tables 3 and 16. The flip probability for Set-LLM is guaranteed to be 0%, and this is confirmed by the empirical results. In contrast, the flip probabilities for the finetuned original models are between 3 and 23%, highlighting that order bias impacts final performance. The flip probabilities are quite consistent across the PIQA, CSQA, and SIQA, although the number of permutations varies between 2 and 24, but the flip probabilities on ARC are significantly higher. Generally, smaller models have higher flip probabilities.

We thank the anonymous reviewer for their suggestion to include flip probabilities as an additional metric.

## E.5 Out-of-distribution performance

Comparison of the out-of-distribution performance finetuned Set-LLM and baseline models. The base model is Gemma 2B.

Table 18: Results on out-of-distribution datasets using Gemma 2B as the base model. In-distribution results are grayed out. All scores are accuracies (%).

| Model | Finetune Dataset | PIQA Std. | PIQA Adv. | ARC Std. | ARC Adv. | CSQA Std. | CSQA Adv.† | SIQA Std. | SIQA Adv. |
|---|---|---|---|---|---|---|---|---|---|
| Pretrained | - | 57.45 | 30.96 | 36.03 | 7.68 | 34.92 | 16.46 | 39.29 | 12.74 |
| Pretrained^Ultra | - | 68.31 | 50.49 | 43.18 | 14.93 | 45.21 | 27.35 | 46.72 | 16.84 |
| Causal Mask+PE^Ultra | | 83.98 | 77.31 | 54.45 | 26.79 | 59.68 | 41.52 | 56.79 | 35.11 |
| + Majority Vote | PIQA | 83.57 | 83.57 | 56.74 | 56.74 | 61.51 | 61.51 | **58.34** | **58.34** |
| SetMask+SetPE^Ultra | | **85.80** | **85.80** | **58.02** | **58.02** | **63.47** | **63.47** | 56.86 | 56.86 |
| Causal Mask+PE^Ultra | | 67.27 | 47.55 | 56.32 | 26.88 | 57.61 | 35.71 | 54.91 | 31.73 |
| + Majority Vote | ARC | 66.92 | 66.92 | 59.56 | 59.56 | 59.38 | 59.38 | 56.86 | 56.86 |
| SetMask+SetPE^Ultra | | **68.61** | **68.61** | 65.02 | 65.02 | **63.39** | **63.39** | **60.64** | **60.64** |
| Causal Mask+PE^Ultra | | 71.84 | 56.64 | 51.63 | 26.45 | 77.89 | 68.47 | 55.72 | 43.14 |
| + Majority Vote | CSQA | **72.03** | **72.03** | 53.58 | 53.58 | 78.46 | 78.46 | 56.81 | 56.81 |
| SetMask+SetPE^Ultra | | 71.49 | 71.49 | **55.38** | **55.38** | 80.18 | 80.18 | **58.96** | **58.96** |
| Causal Mask+PE^Ultra | | 71.55 | 54.68 | 53.52 | 27.30 | 64.76 | 45.86 | 74.33 | 63.97 |
| + Majority Vote | SIQA | 71.33 | 71.33 | 55.55 | 55.55 | 65.85 | 65.85 | 75.23 | 75.23 |
| SetMask+SetPE^Ultra | | **74.16** | **74.16** | **56.83** | **56.83** | **67.73** | **67.73** | 76.15 | 76.15 |

^Ultra Additional pretraining     †Only first 24 (of 120) permutations tested

## E.6 Impact of Set-LLM training to performance on OOD non-set tasks

In addition to testing the out-of-distribution performance on other set-input tasks, we also measure the influence of the Set-LLM finetuning to non-multiple choice questions. Ideally, Set-LLM could still be used for non-set tasks without an impact on performance.

We chose two standard LLM benchmarks to test the effect of set-LLM finetuning on non-multiple-choice questions. For question-answering, we use 1000 test questions from SQuAD v2 [35], and for machine translation, we use the English-German subset from Flores-200 [6]. For SQuAD v2 we record accuracy, and for Flores-200, we record COMET scores.

As finetuned models, we picked models finetuned on PIQA (arbitrarily) and multi-document QA [27]. We test the models in 0-shot, 1-shot, and finetuned settings. We compare the original models with the Set-LLM models. PIQA models were finetuned on multiple-choice questions and therefore trained to copy a choice from the input to the output. This creates a strong bias that is likely to have a significant impact on OOD performance. Multi-document QA models were finetuned to answer a question based on a set of (retrieved) documents.

The results are presented in Tables 19 and 20. The PIQA models were finetuned to copy a choice from the input to the output. We find that these models struggle with SQuAD in the 0-shot and 1-shot (to a lesser extent) settings. The multi-document QA models perform better. Gemma 2B Set-LLM, in particular, performs very poorly. For Flores-200, the differences are less pronounced.

Overall, we find that Gemma 2B SetMask+SetPE^Ultra underperforms Gemma 2B Causal Mask+PE^Ultra in some settings, but not when finetuned. However, the Llama-based models generally do not have a significant gap between original and Set-LLM performance. This is true for both datasets and across almost all evaluation settings. The exception is Llama 3.2 3B finetuned on PIQA in the 0-shot and 1-shot settings.

We conclude that Set-LLM does not have a negative impact on finetuned downstream performance. However, it is more susceptible to overfitting to multiple-choice question answering when trained on such a task.

Table 19: Performance of models finetuned on multiple-choice dataset PIQA applied to standard non-set language tasks. We report accuracy scores for SQuAD v2 and COMET scores for Flores-200 EN→DE in 0-shot, 1-shot, and finetuned settings.

| LLM | Model | SQuAD v2 | | | Flores-200 EN→DE | | |
|---|---|---|---|---|---|---|---|
| | | 0-shot* | 1-shot* | finetuned | 0-shot | 1-shot | finetuned |
| Gemma 2B | Causal Mask+PE$^{\text{Ultra}}$ | 0.57 | 0.62 | 0.89 | 0.76 | 0.73 | 0.71 |
| | SetMask+SetPE$^{\text{Ultra}}$ | 0.25 | 0.46 | 0.91 | 0.66 | 0.66 | 0.68 |
| Llama 3.2 1B | Causal Mask+PE$^{\text{Ultra}}$ | 0.59 | 0.64 | 0.83 | 0.71 | 0.68 | 0.70 |
| | SetMask+SetPE$^{\text{Ultra}}$ | 0.58 | 0.63 | 0.90 | 0.69 | 0.65 | 0.70 |
| Llama 3.2 3B | Causal Mask+PE$^{\text{Ultra}}$ | 0.79 | 0.76 | 0.91 | 0.80 | 0.77 | 0.79 |
| | SetMask+SetPE$^{\text{Ultra}}$ | 0.70 | 0.69 | 0.92 | 0.72 | 0.69 | 0.77 |

*Unanswerable questions were removed for 0-shot and 1-shot settings (SQuAD v2)

Table 20: Performance of models finetuned on Multi-document question answering [27] applied to standard non-set language tasks. We report accuracy scores for SQuAD v2 and COMET scores for Flores-200 EN→DE in 0-shot, 1-shot, and finetuned settings.

| LLM | Model | SQuAD v2 | | | Flores-200 EN→DE | | |
|---|---|---|---|---|---|---|---|
| | | 0-shot* | 1-shot* | finetuned | 0-shot | 1-shot | finetuned |
| Gemma 2B | Causal Mask+PE$^{\text{Ultra}}$ | 0.69 | 0.70 | 0.88 | 0.66 | 0.67 | 0.71 |
| | SetMask+SetPE$^{\text{Ultra}}$ | 0.51 | 0.51 | 0.87 | 0.62 | 0.64 | 0.68 |
| Llama 3.2 1B | Causal Mask+PE$^{\text{Ultra}}$ | 0.75 | 0.74 | 0.84 | 0.69 | 0.65 | 0.69 |
| | SetMask+SetPE$^{\text{Ultra}}$ | 0.75 | 0.72 | 0.90 | 0.71 | 0.68 | 0.69 |
| Llama 3.2 3B | Causal Mask+PE$^{\text{Ultra}}$ | 0.85 | 0.82 | 0.91 | 0.78 | 0.74 | 0.79 |
| | SetMask+SetPE$^{\text{Ultra}}$ | 0.83 | 0.81 | 0.93 | 0.75 | 0.73 | 0.78 |

*Unanswerable questions were removed for 0-shot and 1-shot settings (SQuAD v2)

### E.7 Runtimes & memory usage

Table 21 shows the runtimes and memory usage for Causal Mask+PE and SetMask+SetPE models with Gemma base models. All models were trained and evaluated on Nvidia H200 GPUs on an internal cluster.

The total GPU time for the paper is estimated to be around 1200 hours on Nvidia H200 GPUs.

Table 21: Number of model runs per input, pre-finetuning time on Ultrafeedback, and finetuning and evaluation times on ARC-Challenge. Runtimes are calculated with Gemma 2B as the base model. $k$ is the number of (multiple) choices in the input question. Memory usage is for finetuning on ARC.

| LLM | Model | No. Runs per Input | Pretraining on UltraFeedback (s) | Finetuning on ARC (s) | Memory Usage (GB) | Evaluation on ARC (s) |
|---|---|---|---|---|---|---|
| | Causal Mask+PE | 1 | 6083.50 | 4880.68 | 13.73 | 357.63 |
| Gemma 2B | + Majority Vote | $k!$ | - | - | - | 8620.81 |
| | SetMask+SetPE | 1 | 6002.45 | 4908.22 | 13.61 | 365.47 |
| | Causal Mask+PE | 1 | 9983.51 | 7852.38 | 40.37 | 331.41 |
| Gemma 7B | + Majority Vote | $k!$ | - | - | - | 8448.37 |
| | SetMask+SetPE | 1 | 10003.62 | 7821.80 | 40.02 | 329.77 |

## E.8 Majority vote vs Set-LLM outputs

When calculating the majority vote, the vote count could be considered as a measure of model confidence. In Figures 6 to 9 (left), we compare vote count with accuracy and see that higher vote counts indeed exhibit higher accuracies on average, though the relationship is not as clear-cut for CommonsenseQA.

We explore whether our permutation-invariant model makes similar predictions to the base model, in particular for samples with high vote counts. Figures 6 to 9 (right) shows the agreement rate between the Gemma 2B Causal Mask+PE$^{Ultra}$ baseline and SetMask+SetPE$^{Ultra}$. We see a similar trend to the accuracy, whereby the agreement is higher for high vote count samples. However, the overall agreement rate of (0.6817) on ARC-Challenge demonstrates that the models often make different mistakes. Indeed the agreement rate when Causal Mask+PE$^{Ultra}$ + Majority Vote is incorrect is only around 50% for three of the four datasets.

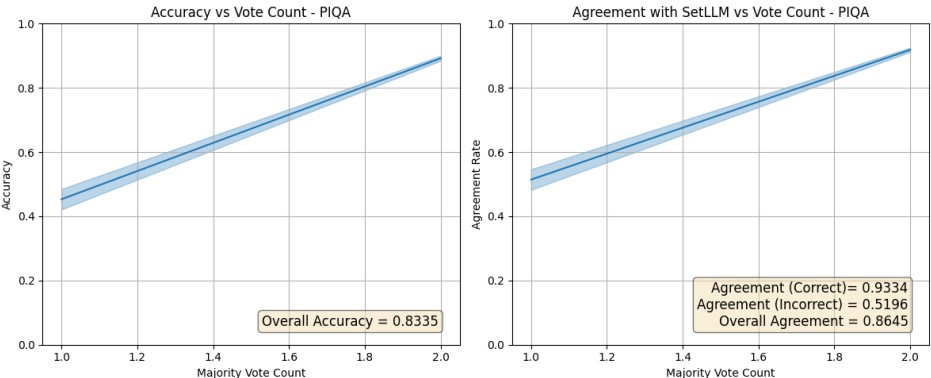

Figure 6: *Causal Mask+PE$^{Ultra}$+Majority Vote* vote count versus accuracy (left). *Causal Mask+PE$^{Ultra}$+Majority Vote* vote count versus agreement rate with Set-LLM (right) on PIQA.

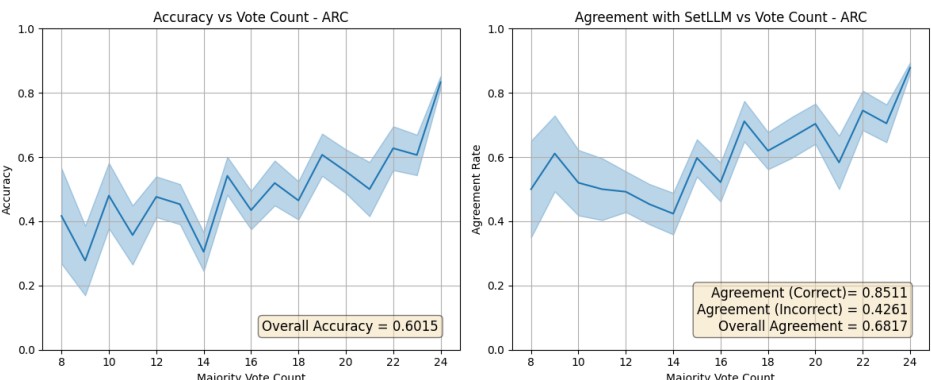

Figure 7: *Causal Mask+PE$^{Ultra}$+Majority Vote* vote count versus accuracy (left). *Causal Mask+PE$^{Ultra}$+Majority Vote* vote count versus agreement rate with Set-LLM (right) on ARC.

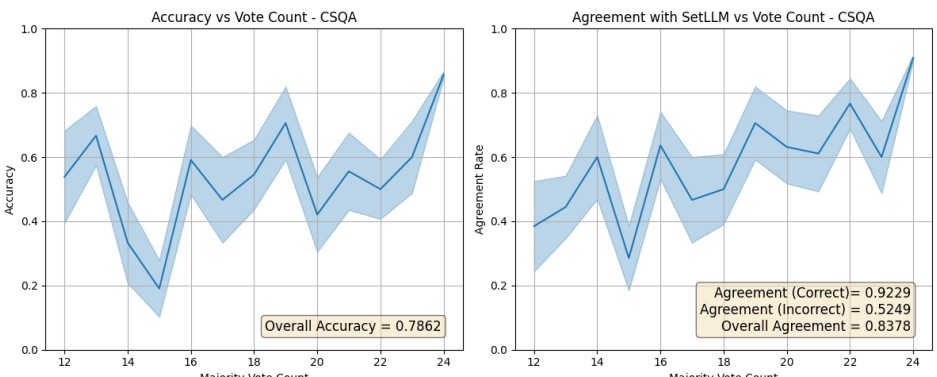

Figure 8: *Causal Mask+PE^Ultra+Majority Vote* vote count versus accuracy (left). *Causal Mask+PE^Ultra+Majority Vote* vote count versus agreement rate with Set-LLM (right) on CSQA.

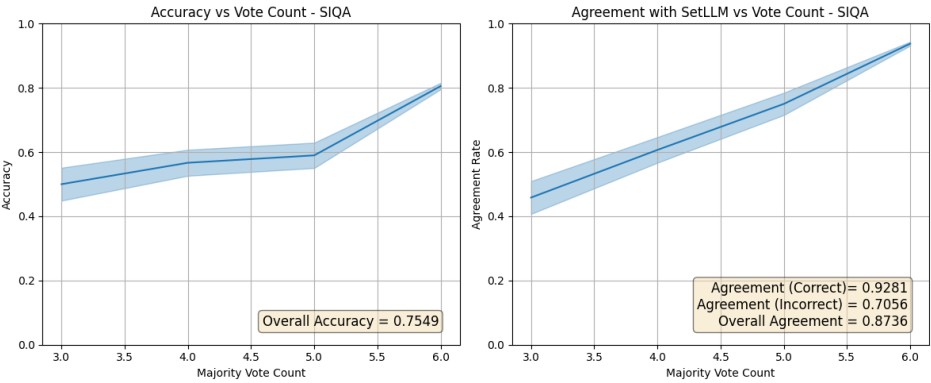

Figure 9: *Causal Mask+PE^Ultra+Majority Vote* vote count versus accuracy (left). *Causal Mask+PE^Ultra+Majority Vote* vote count versus agreement rate with Set-LLM (right) on SIQA.

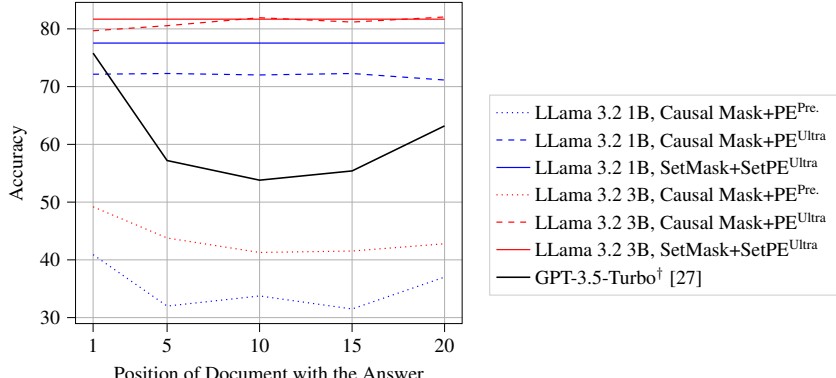

Figure 10: Varying the position of the relevant document (containing the answer) within the pretrained language models' (LLama...$^{\text{Pretrain}}$ and GPT-3.5-Turbo$^{\dagger}$) inputs results in U-shaped performance curves. However, the Set-LLM adapted models produce completely flat performance curves with a higher accuracy. Finetuning the base model results in a mostly flat and similarly high performance curve. $^{\dagger}$GPT-3.5-Turbo was not finetuned on the dataset.

### E.9   Additional multi-document question answering results

The results in Figure 10 reveal a less pronounced U-shaped curve for the pretrained base model (Llama 3.2 3B) than those observed in previous work [27]. The finetuned base model again produces a mostly flat curve with no clear evidence of position bias. The Set-LLM curve is completely flat as expected, and scores slightly higher on average than the finetuned base model.

### E.10   LLM-as-a-judge

We run initial LLM-as-a-judge experiments to showcase the potential of Set-LLM as a permutation-invariant evaluator. We train and test our models on the Search Arena dataset [31]. We restrict the search arena dataset to questions with a single turn (not multiturn), with a winner (no ties), and no longer than 5000 characters. We use a 2/3-1/3 train-test split.

We use the smaller Gemma and LLama models from the rest of the paper. However, it should be noted that LLM evaluators/judges are typically large, powerful models whose output can be used for both evaluation and finetuning. We report accuracy scores with respect to human judgments. The results are shown in Table 22.

We see that in all cases, the Set-LLM model closely matches the baseline model in random-order performance, and significantly outperforms it in the adversarial setting. The performance of the baseline models even drops below random guessing in the adversarial setting. Although the results are promising, a more thorough study should be conducted with realistic LLM evaluator model sizes.

Table 22: LLM-as-a-judge application with different base LLMs. All models were pretrained on UltraFeed-back [7] and then finetuned and tested on Search Arena [31]. All metrics are accuracies to two significant figures.

| LLM | Model | Rand. | Adv. |
|---|---|---|---|
| Rand. Baseline | | 0.50 | 0.50 |
| Gemma 2B | Causal Mask+PE$^{\text{Ultra}}$ | **0.58** | 0.40 |
| | SetMask+SetPE$^{\text{Ultra}}$ | 0.57 | **0.57** |
| Llama 3.2 1B | Causal Mask+PE$^{\text{Ultra}}$ | 0.57 | 0.38 |
| | SetMask+SetPE$^{\text{Ultra}}$ | **0.60** | **0.60** |
| Llama 3.2 3B | Causal Mask+PE$^{\text{Ultra}}$ | **0.62** | 0.47 |
| | SetMask+SetPE$^{\text{Ultra}}$ | **0.62** | **0.62** |

