# OpenReview forum: "Set-LLM: A Permutation-Invariant LLM"
_NeurIPS.cc/2025/Conference — NeurIPS 2025 poster_

### Official Review · Reviewer_f4vF · 2025-06-17

**Clarity:** 3
**Significance:** 2
**Originality:** 2
**Rating:** 5
**Confidence:** 4

**Summary:**

The predictions of LLMs generally depends on the order of the tokens passed as input. While this is generally desirable for sequence modeling, it is undesirable in cases where the LLMs is presented with a set of inputs whose order is not relevant such as LLM-as-a-judge or multiple choice question answering.

This paper presents architecture modifications that allows to fine-tune an LLM so that it can handle tokens whose are arbitrary. Experiments are conducted on multiple-choice formulation benchmarks which shows the proposed method can perform predictions equivariant to the order of set inputs.

**Questions:**

While the paper is very well written and the motivation is relevant, I believe the application is very restricted and the experiments are potentially biased toward emphasizing the problem the paper tackle.

**Restricted application.**

Regarding the application being restricted, you mention LLM-as-a-judge and multiple choice formulation (MCF). However, as you rightly point out, LLMs can be made equivariant by averaging the predictions under all possible permutations. While I agree that it leaves the application useful for MCF which has often 4 choices, the case for LLM-as-a-judge is limited as LLMs just needs to average two options. I think this strongly limits the impact for LLM-as-a-judge as the cost of 1) having a custom position encoding / attention mask 2) requiring a custom fine-tuning is likely to far exceed the the benefit of saving a 2x factor in the cost.

Regarding MCF, I agree though that averaging all 24 permutations would incur substantial extra cost. However, this application is restricted too given that MCF are most often used for evaluations and not much from typical user queries. Furthermore, it is only relevant to benchmarks using MCF and not cloze formulations which further restricts the scope.

In addition to the limited set of applications, the method requires fine-tuning with custom mask and positional embeddings which would also limit the adoption.

**Bias of experiments.**
Regarding the bias of experiments, I believe the paper overemphasises the issue of the order in several ways. First, I believe the metric chosen is a worse case scenario since you pick the one permutation among 24 adversarially. I think this metric does not really show the probability that an inversion occurs (all models will get low adversarial scores when $k!$ becomes large). I would find the results more convincing if you report the fraction of times the predictions differ from the mode as it is probably closer from what a LLM user cares about.

Second, I think the problem you mention of having predictions swapped with the input occur a lot more for small models. This can be seen in Tab 3. for instance where the gap between random and adversarial seems to decay with model size (eg when going from 2B -> 7B and 1B -> 8B). I believe this should be properly analyzed and mentioned in the paper, ideally including a 70B model as the scope of the paper would be reduced if the problem occur just for small models.

I would consider improving my score if they are other applications with a sufficiently large set size so that the method is relevant, if there is a clear evidence that the swapping happens enough and does not vanish to a large extent when using larger models.

**Ethical Concerns:**

["NO or VERY MINOR ethics concerns only"]

**Final Justification:**

I raised my score significantly from 2 to 5. This is because the author addressed my main critics: the potential lack of scope/impact and the way results may be biased.

Regarding the first one, they included experiments on relevant use-cases with much larger set cardinalities where 1) an equivariant approach would indeed be more beneficial (as one cannot average all the k! permutations) 2) significant improvements were shown over the baselines.

Regarding the potential bias, I think the initial (adversarial) metrics chosen by the authors was not very illustrative of what an end-user will care about (eg what is the probability of having a different answer when permuting the output. They mention they will use a better metric and also now discuss clearly the fact that positional bias tends to diminish with larger model which is an important limitation.

**Limitations:**

Yes

**Paper Formatting Concerns:**

* l62: you may want to discuss heads in text if you dont want to incorporate in your notations, they are currently not discussed
* l71: \text{exp} missing
* l106: typo there are N + 1 tokens in q

**Quality:**

3

**Strengths And Weaknesses:**

Strengths:
* The paper is well written and easy to understand
* The experimental setup is clear and well described

Weaknesses:
* The paper targets a niche aspect of LLMs and may have little to no practical use
* In addition to having a potentially limited impact, the solution proposed by the author appears very simple (modify the positional encoding and the attention mask) and requires fine-tuning the model
* The experiments while being correct are biased towards the result the author want to show

---

> ### Author Response · Authors · 2025-08-01
> **Rebuttal**
>
> We sincerely thank Reviewer f4vF for the careful review and the thought-provoking questions that help us to improve our paper.
>
> > W1: The paper targets a niche aspect of LLMs and may have little to no practical use.
>
> Respectfully, we do not agree with the reviewer that this is a niche aspect of LLMs with little to no practical use. Many works identify order bias as a problem for LLMs, and we present an effective, theoretically grounded solution.
> While we chose Multiple-choice QA for benchmarking, we emphasize that our method extends to other LLM tasks, such as LLM as a judge, multi-document summarization, and retrieval-augmented generation. To strengthen our paper, we have conducted additional experiments on these tasks. The results validate the general applicability and benefits of the proposed method. We will expand these and include them in the final paper.
>
> **Multi-Document Summarization**
>
> We conduct our evaluation on the MultiNews [1] dataset. We filter out inputs of length greater than 20000 characters and use the standard train-validation-test split. We compare the original model architecture with Set-LLM with single inference runs (and no adversarial setting). We use the best-performing hyperparameters from the multiple-choice experiments.
>
> We report standard metrics: Rouge F1 scores and compression (the length of the original text divided by the length of the summary). We also include Rouge F1 scores between the target summaries and the model summaries (R1*, R2*, R3*).
>
> |Model             |Variant        |Compression |R1      |R2      |RL      |R1*      |R2*     |RL*    |
> |-------------------|----------------|-------------------|---------|---------|---------|---------|---------|---------|
> |Target             |                    |5.18               |0.30    |0.14    |0.16    | -        | -         | -        |
> |Gemma 2B     |Original       |13.21             |0.19    |0.14    |0.15    |0.32    |0.10    |0.19   |
> |                       |Set-LLM      |6.39               |0.30    |0.23    |0.23    |0.48    |0.19    |0.26   |
> |Gemma 7B     |Original       |8.11               |0.27    |0.21    |0.22    |0.42    |0.15    |0.23   |
> |                       |Set-LLM      |6.64               |0.30    |0.23    |0.23    |0.50    |0.22    |0.28   |
> |Llama 3.2 3B  |Original       |8.86               |0.25    |0.17    |0.18    |0.42    |0.14    |0.22   |
> |                       |Set-LLM      |6.31               |0.30    |0.20    |0.19    |0.50    |0.20    |0.26   |
>
> Set-LLM outperforms the finetuned baseline with all three base models in all metrics. However, the margin is significantly narrower with the larger Gemma 7B model. Note that the compression rate is always lower for Set-LLM, meaning the summaries it produces are longer.
>
> [1] Fabbri, Alexander R., et al. "Multi-news: A large-scale multi-document summarization dataset and abstractive hierarchical model." arXiv preprint arXiv:1906.01749 (2019).
>
>
> **Multi-Document Question Answering (Retrieval-Augmented Generation)**
>
> Motivated by [2], we evaluate Set-LLM on retrieval-augmented question answering. [2] introduces a task mimicking the RAG setup. They ask questions with 20 supporting documents, where only one document contains the information required for the answer. They show that LLMs (e.g., GPT3.5 Turbo) produce a U-shaped performance curve given the location of the answer. That is, if the document with the answer is listed first or last, then the LLM answers the question correctly with a much higher probability (e.g., 75% in position 1/20 vs 55% in position 10/20). In contrast, Set-LLM is guaranteed to have a flat performance curve, since reordering the documents does not affect the outcome.
>
> We split the dataset into 60% training, 10% validation, and 30% test data. We replicate the training data 5 times and shuffle the documents randomly for each input. This is done to avoid the model overfitting to a specific formulation of each training question. Again, hyperparameters were taken from the multiple-choice experiments.
>
> We have 5 different test sets with the answer document placed in the first, 5th, 10th, 15th, and last positions. We provide results for two LLama-based models below. All scores denote percentage accuracy.
>
> |Model             |Variant        |1         | 5        | 10     | 15      | 20      |
> |-------------------|----------------|---------|---------|---------|---------|---------|
> |Llama 3.2 1B  |Original      |34.76  |26.85  |25.22  |28.23  |37.39  |
> |                       |Set-LLM     |76.54  |76.54  |76.54  |76.54  |76.54  |
> |Llama 3.2 3B  |Original      |54.83  |44.29  |44.92  |44.04  |52.07  |
> |                       |Set-LLM     |81.68  |81.68  |81.68  |81.68  |81.68  |
>
> [2] Nelson F. Liu, Kevin Lin, John Hewitt, Ashwin Paranjape, Michele Bevilacqua, Fabio Petroni, and Percy Liang. 2024. Lost in the Middle: How Language Models Use Long Contexts. Transactions of the Association for Computational Linguistics, 12:157–173.

---

> > ### Author Response · Authors · 2025-08-01
> >
> > > W2: In addition to having a potentially limited impact, the solution proposed by the author appears very simple (modify the positional encoding and the attention mask) and requires fine-tuning the model
> >
> > The solution does not require any complex changes to the architecture, yet it is effective, which we consider a significant advantage of the proposed method.
> >
> > With regards to the finetuning. There is indeed an initial cost associated with the finetuning. However, first, the amount of fine-tuning (4h on a single GPU) is negligible in comparison to the overall pre-training time (e.g., 1.46M GPU hours for llama 3.1 8B), and second, finetuning is not required for every downstream task.
> >
> > The main purpose of finetuning is to allow the model to adapt to its new, altered architecture. We envisage that a typical scenario would involve finetuning Set-LLM on a general-purpose set-dataset and then using it for relevant downstream applications. For the multiple-choice datasets, for example, we provide results for training on one dataset and then testing on another in Table 16 in the appendix. Set-LLM consistently outperforms the base model in this out-of-distribution setting.
> >
> > Furthermore, the only alternative for overcoming order bias that does not require finetuning is aggregating all permutations. This carries a higher inference-time cost and higher total cost than our fine tuning, even with these small test sets (please refer to Table 17 in the appendix for runtimes).
> >
> > > W3: LLM-as-a-Judge usually only has 2 options ...requiring a custom fine-tuning is likely to far exceed the benefit of saving a 2x factor in the cost
> >
> > We would argue that even for the case of two options, a reduction of inference cost by a factor of two is already significant.  As reviewer nn3i mentions “the computational overhead seems manageable”, finetuning only requires 4h on a single GPU. Arguably, this leads to an amortization of the finetuning costs already after 8h of inference time of an LLM as a judge. And though less common, multiple-choice LLM judges are also used [3].
> >
> > Additionally, the proposed approach avoids ties, which can easily occur with two options - when option A is ranked better than option B in one case and option B better than option A in the other case.
> >
> > [3] Gu, Jiawei, et al. "A survey on llm-as-a-judge." arXiv preprint arXiv:2411.15594 (2024).
> >
> > > W4: The experiments while being correct are biased towards the result the authors want to show. I would find the results more convincing if you report the fraction of times the predictions differ from the mode.
> >
> > Please note that we are already reporting the average case performance (i.e. random permutation performance), which is an unbiased estimator for the average prediction accuracy over all permutations. We do not believe that this is a biased setup of the experiments.
> >
> > However, the average case performance does not tell the full story, it hides the order bias. Although order bias averages out over a large dataset with random permutations, this is not the case for individual samples, and this explains why there is research interest in this issue. The point of the adversarial setting isn’t to show how bad the model is, but rather how sensitive it is to order (and what the worst case impact of that could be).
> >
> > Moreover, with the additional retrieval-augmented generation results, we can see that even average case performance can significantly improve by making the LLM order-invariant. An improvement here is significant, and is likely due to the large number of documents (20) with the potential for individual documents to be “lost in the middle” [4]
> >
> > [4] Nelson F. Liu, Kevin Lin, John Hewitt, Ashwin Paranjape, Michele Bevilacqua, Fabio Petroni, and Percy Liang. 2024. Lost in the Middle: How Language Models Use Long Contexts. Transactions of the Association for Computational Linguistics, 12:157–173.
> >
> >
> > > W5:  I think the problem you mention of having predictions swapped with the input occur a lot more for small models. …Clear evidence that the swapping happens often enough and does not vanish for larger models.
> >
> > Indeed the swapping decreases as the model size increases in our experiments, and this trend continues to some degree as the model size is further increased. However the drops in performance for large models are still very significant. This can be seen in [5]: For Llama2-70B, there are still drops of 20-30% on the datasets we use; and even GPT-3.5-turbo sees drops between 6-25%.
> >
> > We have decided not to rerun these experiments, because they are rather costly and the results are unlikely to differ significantly. Admittedly, these are previous generation models, but the newer (small) models we use show the same trend.
> >
> > [5] Zong, Yongshuo, et al. "Fool your (vision and) language model with embarrassingly simple permutations." arXiv preprint arXiv:2310.01651 (2023).

---

> > > ### Comment · Reviewer_f4vF · 2025-08-04
> > > **Answer to rebuttal**
> > >
> > > Thank you for your thorough answer.
> > >
> > > * I acknowledge your experiments which are supporting use-cases where beyond cases with 2 or 4 choices which makes the argument more compelling and the paper more impactful
> > > * Regarding fine-tuning, I believe this is still an extra cost (both in complexity and operational overhead) but it is not a major point
> > > * Regarding the fact that the bias is less significant for bigger models, I understand that you do not want to rerun all your experiments. This point has to be raised though as a potential limitation. The point you make about Llama2-70B is true but it is also because you are using an old model, this bias will be lower for Llama3-70B for instance.
> > >
> > > > Please note that we are already reporting the average case performance (i.e. random permutation performance), which is an unbiased estimator for the average prediction accuracy over all permutations. We do not believe that this is a biased setup of the experiments.
> > >
> > > I do not think random permutation performance is showing much about the fraction of times the prediction "flips". I highly encourage the author to report a less biased metric such as the fraction of times the predictions differ from the mode. As noted in my review, when the size of the set is large, the adversarial metric may become bad because of the noise as only one flip has to occur against $O(k!)$. Even a good model that is equivariant in 99.9% of cases would have a bad metric for a large $k$ in this setting.

---

> > > > ### Author Response · Authors · 2025-08-04
> > > > **Flip Probability Scores**
> > > >
> > > > Thank you for your valuable feedback.
> > > >
> > > > Regarding Llama3-70B, we don’t think we can provide any new results during the discussion period, but we are happy to add a remark to the limitations and will endeavour to include such results in a final version of the paper.
> > > >
> > > > Regarding the suggested metric, we have calculated the “flip probability” for our models. We provide results for all models from the main table of results (Table 3). The models are identical to those in Table 3, i.e., “Original” below means “Causal Mask+PE+Ultra”. Please note that the flip probability for Set-LLM is guaranteed to be 0%, and this is confirmed by the empirical results. In contrast the flip probabilities for the finetuned original models are between 3 and 23%, highlighting that order bias impacts final performance.
> > > >
> > > > |LLM                |Model         |PIQA  |ARC   |CSQA |SIQA  |
> > > > |-------------------|----------------|---------|---------|---------|---------|
> > > > |Gemma 2B     |Original      |  6.66  |21.45  |  5.40  |  6.35  |
> > > > |                       |Set-LLM     |  0.00  |  0.00  |  0.00  |  0.00  |
> > > > |Gemma 7B     |Original      |  3.37  |  7.92  |  2.94  |  4.32  |
> > > > |                       |Set-LLM     |  0.00  |  0.00  |  0.00  |  0.00  |
> > > > |Llama 3.2 1B  |Original      |  8.24  |23.37  |  5.72  |  5.70  |
> > > > |                       |Set-LLM     |  0.00  |  0.00  |  0.00  |  0.00  |
> > > > |Llama 3.2 3B  |Original      |  5.20  |11.76  |  3.32  |  4.39  |
> > > > |                       |Set-LLM     |  0.00  |  0.00  |  0.00  |  0.00  |
> > > > |Llama 3.1 8B  |Original      |  4.52  |  7.87  |  3.23  |  4.63  |
> > > > |                       |Set-LLM     |  0.00  |  0.00  |  0.00  |  0.00  |
> > > >
> > > > We are happy to add these results to the final paper to supplement the other metrics, along with an acknowledgement to the anonymous reviewer for the suggestion. Please let us know if there are any other scores you would like us to provide.

---

> > > > > ### Comment · Reviewer_f4vF · 2025-08-04
> > > > > **Answer**
> > > > >
> > > > > Thank you very much, this analysis is very helpful.
> > > > >
> > > > > Between the Llama series in particular, it does really show the gap going to zero when model size increases. I think mentioning this, and also include the flip probability for Llama3-70B would make one potential limitation of the paper clearer (there may not be that many flips when using sufficiently advanced model, and this number may be going down overtime) and also report results in a way that is closer to what an end user will care about (how likely are flips rather than whether a flip occur given a large set of sample when k! is large).
> > > > >
> > > > > That being said, I do not think this analysis showing how the gap decreases with "model capacity" removes quality on the paper but on the contrary improves it by being clear on one of its potential limitation. With the experiments previously shared that increased the scope a lot by considering larger "sets", I think the paper quality increased significantly and I will raise my score to accept to reflect this.

---

> > > > > > ### Author Response · Authors · 2025-08-04
> > > > > >
> > > > > > Thank you for your continued engagement and for helping us improve the paper. We appreciate you increasing your score. Somehow, the old score has disappeared, and we do not see any new score at the moment, but that might just be a bug in the system.

---

> > > > > > > ### Comment · Reviewer_f4vF · 2025-08-05
> > > > > > > **answer**
> > > > > > >
> > > > > > > Do not worry, this is not a bug, see author instructions:
> > > > > > >
> > > > > > > > Author-reviewer discussion will be closed on August 6 11:59pm AoE. As reviewers update “Final Justification” and “Rating”, this information will not be visible to authors until the final paper decisions are out.

---

### Official Review · Reviewer_nn3i · 2025-06-18

**Clarity:** 4
**Significance:** 3
**Originality:** 3
**Rating:** 5
**Confidence:** 3

**Summary:**

In this work, the authors present Set-LLM, a novel architectural modification to pretrained large language models designed to mitigate their sensitivity to input order. Specifically, they propose changes to the positional encoding and attention masking mechanisms to enable the model to handle mixed set-text inputs—such as text sequences containing permutable components like option lists—while ensuring permutation invariance. They evaluate their proposed dataset on various MCQ benchmarks and with different models, ranging from 1B to 8B.

**Questions:**

- Concerning the Set-Mask, I understand that the permutable options should not attend to one another, but why change the causal nature of the mask as well, especially since that can cause severe changes to the model behaviour ?
- There seems to be a missing notation in the statement of Theorem 1, what is \tilde{X}(t+1) ?
- In Table 2, it is surprising that the PrefixMask + No positional embeddings configuration still yields meaningful outputs. Without positional information, the model essentially becomes a bag-of-words processor. Could the authors explain how such a setup maintains coherent output?
- Concerning the adversarial order: do the authors generate multiple completions per input order, or just one? If only a single output is used per permutation, then it becomes difficult to disentangle the effects of input order from the inherent randomness of LLM outputs. Even with a fixed order, generating n samples will eventually yield some incorrect outputs due to sampling variance. In this context, changing the order could function similarly to changing the random seed, complicating the interpretation of the adversarial metric.

**Ethical Concerns:**

["NO or VERY MINOR ethics concerns only"]

**Final Justification:**

This paper proposes Set-LLM, an architectural modification to improve permutation invariance in LLMs for mixed set-text inputs. The problem is timely and well-motivated, and the approach is clearly presented with solid empirical results across multiple models and benchmarks. While the required fine-tuning introduces additional training cost, the overhead seems manageable, and the authors’ additional experiments confirm minimal degradation on general benchmarks. Some methodological choices, such as removing causal masking, could be further explored, but overall, the work offers a meaningful and well-supported contribution.

**Limitations:**

See above

**Quality:**

3

**Strengths And Weaknesses:**

Overall, I found this paper interesting and well-written. The exposition is clear, making the work easy to read and follow.

Relevance:
The problem of order sensitivity in large language models is timely and important. The proposed approach addresses the issue in a meaningful way. The experimental section is convincing, leveraging multiple models and benchmarks to support the claims. However, I do have a few remarks and questions (detailed below).

Limitations:
- A potential limitation is that the proposed modifications to the attention mask and positional embeddings require an additional training step. That said, the computational overhead seems manageable (reported as up to 4 hours on a single H200 GPU for 8B models).
- Switching from a causal mask to a prefix mask could significantly affect model behavior. While this may not be apparent in tasks like multiple-choice questions (MCQs), where outputs are short and structured. It would strengthen the paper to verify that general model capabilities do not degrade on standard benchmarks.
- On a side note, I am not entirely convinced the theorems qualify as such, especially the second one, which seems rather straightforward.

---

> ### Author Rebuttal · Authors · 2025-07-31
>
> We sincerely thank Reviewer nn3i for the positive feedback and thoughtful suggestions.
>
> > W1: Requires finetuning. That said, the computational overhead seems manageable.
>
> We agree with the reviewer, that the proposed fine-tuning method results in an additional training step, however the amount of fine-tuning (4h on a single GPU) is negligible in comparison to the overall pre-training time (e.g. 1.46M GPU hours for llama 3.1 8B) and the resulting model can be used in a range of applications.
>
> > W2: It would strengthen the paper to verify that general model capabilities do not degrade on standard benchmarks.
>
> We endeavour to provide an evaluation of the models after fine tuning on standard evaluation benchmarks they were built for during the author reviewer discussion period. We have machine translation and math in mind, but would be happy to take suggestions.
> Note, that in response to other reviewers, we have evaluated Set-LLM on additional tasks (multi-document summarization and retrieval-augmented generation).
>
> > Q1: Why change the causal nature of the mask?
>
> This is an interesting suggestion. The overall causal structure of the mask has to be destroyed in order to guarantee invariance. However, the attention within an option could remain causal. We have not tried this, but it would be an interesting addition to our ablation studies. We do not expect significant performance gains since a fully-connect prefix mask is shown to be at least as effective as a causal mask in [1].
>
> [1] Kopiczko, Dawid J., Tijmen Blankevoort, and Yuki M. Asano. "Bitune: Bidirectional instruction-tuning." arXiv preprint arXiv:2405.14862 (2024).
>
> > Q2: There seems to be a missing notation in the statement of Theorem 1, what is \tilde{X}(t+1) ?
>
> Thank you for bringing this to our attention. \tilde denotes variables after permuting the input. We will make sure to add this clarification.
>
> > Q3: In Table 2, it is surprising that the PrefixMask + No positional embeddings configuration still yields meaningful outputs.
>
> Indeed, this is very surprising, that a bag-of-words, or in this case better a bag-of-tokens model performs so well, especially the 74.54% on PIQA. This probably says more about the dataset than about the model. Note that the performance on the other datasets is weaker.
>
> > Q4: Concerning the adversarial order: do the authors generate multiple completions per input order, or just one?
>
> We do not generate completions as answers, instead, the possible options are ranked by their log probabilities. Therefore the evaluation is deterministic once the model has been trained.

---

> > ### Author Response · Authors · 2025-08-08
> > **Update on general model capabilities**
> >
> > We would like to provide an update on
> >
> > >W2: It would strengthen the paper to verify that general model capabilities do not degrade on standard benchmarks.
> >
> > We have run experiments on two standard LLM benchmarks: SQuAD for question answering, and Flores-200 for machine translation. We run experiments in 0-shot, 1-shot, and finetuning setups. Overall, we find very few gaps between the original model's performance and the Set-LLM model's performance. Specifically, though, the Gemma 2B Set-LLM model seems to be susceptible to overfitting to the multiple-choice task, but the performance gap disappears after finetuning.
> >
> > Reviewer rzD4 had a similar question, and to avoid duplication, we would like to refer the reviewer to our comment https://openreview.net/forum?id=Eyis2h3tba&noteId=coKHbqo9IA for more details.
> >
> > We hope we have addressed your concerns. Please let us know if you have any further questions.

---

### Official Review · Reviewer_ryD4 · 2025-07-02

**Clarity:** 3
**Significance:** 2
**Originality:** 3
**Rating:** 4
**Confidence:** 5

**Summary:**

This paper proposed set-llm, a technique that can help finetune base llm to be permutation invariant. The key method is here to introduce a new attention masking and a new set position embedding.
Experiments are mainly conducted on multiple choice evaluations with random and adversarial permutations of the choices to test the robustness of the proposed method to permutated choices.
The results suggest that the proposed method indeed is robust to different permutations.

**Questions:**

See weakness above.
I feel that the weakness overweights the strength for this work.

**Ethical Concerns:**

["NO or VERY MINOR ethics concerns only"]

**Final Justification:**

I would raise my score to 4 considering the additional experiments demonstrated. I still have reseverations on the applicability to wider applications, but I think this paper at least should be interesting when considering for specific applications like LLM-as-a-judge.

**Limitations:**

The paper has discussed the limitations

**Quality:**

3

**Strengths And Weaknesses:**

Strength

The overall approach towards the solution for permutation invariant models is straightforward and effective.
The experiments agree with the theory behind this work.
The paper is well written and includes enough ablation studies to test the performance of the proposed model.



Weakness

The scope for this work seems to be rather limited, only targeting the multiple choice QA questions. Consider the method requires finetuning to the base model, I would question if this is necessary to finetune a model just to make it work for multi-choice questions.

Additionally, the paper has shown that the proposed finetuning is able to improve on multiple choice question, I wonder what's the influence of the finetuning process to non-multiple choice questions? Such as code generation tasks measured by HumanEval or other benchmarks.

---

> ### Author Rebuttal · Authors · 2025-07-31
>
> We sincerely thank Reviewer ryD4 for their feedback and for helping us to improve our paper.
>
>
> > W1: The scope for this work seems to be rather limited.
>
>
> We understand the concern. We argue in the paper that the multiple choice setting is a reasonable test case for LLM-as-a-Judge applications, and LLM-as-a-Judge has wide-ranging applications. We mention other possible applications in the paper and see our model as a possible blueprint for these applications.
> To strengthen our paper and showcase the versatility of our approach we have conducted additional experiments on multi document summarization and retrieval augmented generation, both common LLM tasks. The results validate the general applicability and benefits of the proposed method. We have not been able to run all the baseline models on both of these applications, but we will make sure to complete these results for the final version of the paper.
>
>
> To strengthen our paper and showcase the versatility of our approach we have conducted additional experiments on multi document summarization and retrieval augmented generation, both common LLM tasks. The results validate the general applicability and benefits of the proposed method. We have not been able to run all the baseline models on both of these applications, but we will make sure to complete these results for the final version of the paper.
>
> ## Multi Document Summarization
>
> We conduct our evaluation on the MultiNews [1] dataset. We filter out inputs of length greater than 20000 characters to satisfy our memory constraints. We use the standard train-validation-test split. We compare the original model architecture with Set-LLM with single runs (no adversarial setting). Our initial results do not include hyperparameter tuning, we use the best-performing hyperparameters from the multiple-choice experiments. The experiments will be expanded for the final paper.
>
> We report standard metrics, this includes Rouge F1 scores and compression - the length of the original text divided by the length of the summary. We also include Rouge F1 scores between the target summaries and the model summaries (R1*, R2*, R3*).
>
> |Model             |Variant        |Compression |R1      |R2      |RL      |R1*      |R2*     |RL*    |
> |-------------------|----------------|-------------------|---------|---------|---------|---------|---------|---------|
> |Target             |                    |5.18               |0.30    |0.14    |0.16    | -        | -         | -        |
> |Gemma 2B     |Original       |13.21             |0.19    |0.14    |0.15    |0.32    |0.10    |0.19   |
> |                       |Set-LLM      |6.39               |0.30    |0.23    |0.23    |0.48    |0.19    |0.26   |
> |Gemma 7B     |Original       |8.11               |0.27    |0.21    |0.22    |0.42    |0.15    |0.23   |
> |                       |Set-LLM      |6.64               |0.30    |0.23    |0.23    |0.50    |0.22    |0.28   |
> |Llama 3.2 3B  |Original       |8.86               |0.25    |0.17    |0.18    |0.42    |0.14    |0.22   |
> |                       |Set-LLM      |6.31               |0.30    |0.20    |0.19    |0.50    |0.20    |0.26   |
>
> Set-LLM outperforms the finetuned baseline with all three base models in all metrics. However, the margin is significantly narrower with the larger Gemma 7B model. Note that the compression rate is always lower for Set-LLM, meaning the summaries it produces are longer.
>
> [1] Fabbri, Alexander R., et al. "Multi-news: A large-scale multi-document summarization dataset and abstractive hierarchical model." arXiv preprint arXiv:1906.01749 (2019).
>
>
> ## Multi-Document Question Answering (Retrieval-Augmented Generation)
>
> Motivated by [2], we evaluate Set-LLM on retrieval-augmented question answering. The authors introduce a task mimicking the retrieval-augmented generation setup. They ask questions with 20 supporting documents, where only one document contains the information required for the answer. They show that commercial LLMs (GPT3.5 Turbo) produce a U-shaped performance graph given the location of the answer. If the document with the answer is listed first or last, then the LLM answers the question correctly with a much higher probability (e.g. 75% in position 1/20 vs 55% in position 10/20). In contrast, Set-LLM is guaranteed to have a flat performance curve, since reordering the documents does not affect the outcome.
>
> We split the dataset into 60% training, 10% validation, and 30% test data. We replicate the training data 5 times and shuffle the documents randomly for each input. This is done to avoid the model overfitting to a specific formulation of each training question. We have not conducted hyperparameter tuning yet, the hyperparameters were taken from the multiple-choice experiments.
>
> We have 5 different test sets with the answer document placed in 5 different locations (first, 5th, 10th, 15th, last). We provide the results for two LLama-based models below. All scores denote the percentage accuracy.
>
> |Model             |Variant        |1         | 5        | 10     | 15      | 20      |
> |-------------------|----------------|---------|---------|---------|---------|---------|
> |Llama 3.2 1B  |Original      |34.76  |26.85  |25.22  |28.23  |37.39  |
> |                       |Set-LLM     |76.54  |76.54  |76.54  |76.54  |76.54  |
> |Llama 3.2 3B  |Original      |54.83  |44.29  |44.92  |44.04  |52.07  |
> |                       |Set-LLM     |81.68  |81.68  |81.68  |81.68  |81.68  |
>
> [2] Nelson F. Liu, Kevin Lin, John Hewitt, Ashwin Paranjape, Michele Bevilacqua, Fabio Petroni, and Percy Liang. 2024. Lost in the Middle: How Language Models Use Long Contexts. Transactions of the Association for Computational Linguistics, 12:157–173.
>
>
>
>
> > W2: I would question if this is necessary to finetune a model just to make it work for multi-choice questions.
>
>
> To add to the response to W1, the finetuning cost is fairly small (between 1 and 4 GPU hours) for each dataset and model we consider. This is negligible in comparison to the millions of GPU hours used to pretrain these models (e.g. 1.46M GPU hours for llama 3.1 8B). Moreover, the main purpose of finetuning is to allow the model to adapt to its new, altered architecture. We envisage that a typical scenario would involve finetuning Set-LLM on a general-purpose set-dataset and then using it for relevant downstream applications.

---

> > ### Comment · Reviewer_ryD4 · 2025-08-04
> >
> > I would like to thank the author for the rebuttal.
> > However, the rebuttal does not clear my concern on multiple choice QA questions, the rebuttal mentioned that this could be useful for LLM-as-a-judge scenarios, then why not evaluate it on benchmarks like search arena where the evaluation is on LLM-as-a-judge?
> > Additionally, one of my concern is omitted from the response, I am interesting in knowing the influence of the set-llm finetuning to non-multiple choice questions.

---

> ### Author Response · Authors · 2025-08-08
> **non-multiple choice questions**
>
> > I am interesting in knowing the influence of the set-llm finetuning to non-multiple choice questions.
>
> This is a great suggestion! It took some time to get results.
>
> We chose two standard LLM benchmarks to test the effect of set-LLM finetuning on non-multiple-choice questions. For question-answering, we use 1000 test questions from SQuAD v2 [1], and for machine translation, we use the English-German subset from Flores-200 [2]. For SQuAD v2 we record accuracy, and for Flores-200, we record COMET scores.
>
> As finetuned models, we picked models finetuned on PIQA (arbitrarily) and Long Context QA [3] (from the rebuttal). We test the models in 0-shot, 1-shot, and finetuned settings. We compare the original models with the Set-LLM models. PIQA models were finetuned on multiple-choice questions and therefore trained to copy a choice from the input to the output. Long QA models were finetuned to answer a question based on a set of (retrieved) documents.
>
>
> **SQuAD v2 (PIQA)**
>
> |Model             |Variant        | 0-Shot*    | 1-Shot*   | finetuned  |
> |-------------------|----------------|--------------|--------------|--------------|
> |Gemma 2B     |Original       | 0.57        | 0.62         | 0.89         |
> |                       |Set-LLM      | 0.25        | 0.46         | 0.91         |
> |Llama 3.2 1B  |Original       | 0.59       | 0.64          | 0.83         |
> |                       |Set-LLM      | 0.58       | 0.63          | 0.90         |
> |Llama 3.2 3B  |Original       | 0.79       | 0.76          | 0.91         |
> |                       |Set-LLM      | 0.70       | 0.69          | 0.92         |
>
> *Unanswerable questions were removed for 0-shot and 1-shot settings
>
> **SQuAD v2 (LongQA)**
>
> |Model             |Variant        | 0-Shot*    | 1-Shot*   | finetuned |
> |-------------------|----------------|--------------|--------------|--------------|
> |Gemma 2B     |Original       | 0.69        | 0.70         | 0.88         |
> |                       |Set-LLM      | 0.51        | 0.51         | 0.87        |
> |Llama 3.2 1B  |Original       | 0.75        | 0.74         | 0.84         |
> |                       |Set-LLM      | 0.75        | 0.72         | 0.90         |
> |Llama 3.2 3B  |Original       | 0.85        | 0.82         | 0.91         |
> |                       |Set-LLM      | 0.83        | 0.81         | 0.93         |
>
> *Unanswerable questions were removed for 0-shot and 1-shot settings
>
>
> **Flores-200, English -> German (PIQA)**
>
> |Model             |Variant        | 0-Shot     | 1-Shot     | finetuned |
> |-------------------|----------------|--------------|--------------|--------------|
> |Gemma 2B     |Original       | 0.76    | 0.73      | 0.71     |
> |                       |Set-LLM      | 0.66    | 0.66     | 0.68    |
> |Llama 3.2 1B  |Original       | 0.71    | 0.68      | 0.70     |
> |                       |Set-LLM      | 0.69    | 0.65      | 0.70     |
> |Llama 3.2 3B  |Original       | 0.80    | 0.77     | 0.79    |
> |                       |Set-LLM      | 0.72    | 0.69      | 0.77     |
>
> **Flores-200, English -> German (LongQA)**
>
> |Model             |Variant        | 0-Shot     | 1-Shot     | finetuned |
> |-------------------|----------------|--------------|--------------|--------------|
> |Gemma 2B     |Original       | 0.66    | 0.67      | 0.71     |
> |                       |Set-LLM      | 0.62    | 0.64     | 0.68    |
> |Llama 3.2 1B  |Original       | 0.69    | 0.65      | 0.69     |
> |                       |Set-LLM      | 0.71    | 0.68      | 0.69     |
> |Llama 3.2 3B  |Original       | 0.78    | 0.74     | 0.79    |
> |                       |Set-LLM      | 0.75    | 0.73      | 0.78    |
>
>
> The PIQA models were finetuned to copy a choice from the input to the output. We find that these models struggle with SQuAD in the 0-shot and 1-shot (to a lesser extent) settings. The LongQA models perform better. Gemma 2B Set-LLM, in particular, performs very poorly.
> For Flores-200, the differences are less pronounced.
>
> Overall, we find that Gemma 2B Set-LLM underperforms Gemma 2B Original in some settings, but not when finetuned.
>
> However, the Llama-based models generally do not have a significant gap between original and Set-LLM performance. This is true for both datasets and across almost all evaluation settings. The exception is Llama 3.2 3B finetuned on PIQA in 0-shot and 1-shot settings.
>
> [1] Rajpurkar, Pranav, Robin Jia, and Percy Liang. "Know what you don't know: Unanswerable questions for SQuAD." arXiv preprint arXiv:1806.03822 (2018).
>
> [2] Costa-Jussà, Marta R., et al. "No language left behind: Scaling human-centered machine translation." arXiv preprint arXiv:2207.04672 (2022).
>
> [3] Nelson F. Liu, Kevin Lin, John Hewitt, Ashwin Paranjape, Michele Bevilacqua, Fabio Petroni, and Percy Liang. 2024. Lost in the Middle: How Language Models Use Long Contexts. Transactions of the Association for Computational Linguistics, 12:157–173.

---

> ### Author Response · Authors · 2025-08-08
> **Search Arena**
>
> >Why not evaluate on benchmarks like search arena where the evaluation is on LLM-as-a-judge?
>
> The main aim of this paper is to create a set-invariant LLM architecture using pretrained LLMs. This then has several potential applications as discussed in the paper and the rebuttal. We believe that overall, the experiments convincingly show the viability of the approach. The aim of the paper is not to explore in detail the specific application of LLM as a Judge, though admittedly this may not have been clear in the initial submission, especially with the focus of the experiments on a single task. However, with the help of the feedback from the reviewers and the additional results, we have expanded the scope of the experimental section. As such, we feel a deep-dive into Set-LLM for LLM as a Judge (or other applications) should be deferred to future work. We have run some initial experiments on the suggested search arena dataset, but we have used the small pretrained LLMs from the paper, and the results are not conclusive.
>
> **Search Arena (finetuned)**
>
> |Model             |Variant        |rand. Order | adv. Order  | flip prob.*    |
> |-------------------|----------------|----------------|----------------|----------------|
> |Rand. Baseline |                | 0.50           | 0.50             | 0.25           |
> |Gemma 2B     |Original      | 0.58           | 0.40             | 0.18           |
> |                       |Set-LLM     | 0.57           | 0.57            | 0.00           |
> |Llama 3.2 1B  |Original      | 0.57           | 0.38             | 0.19           |
> |                       |Set-LLM     | 0.60           | 0.60             | 0.00           |
> |Llama 3.2 3B  |Original      | 0.62           | 0.47            | 0.15           |
> |                       |Set-LLM     | 0.62           | 0.62             | 0.00           |
>
> *As suggested by Reviewer f4vF
>
> We restricted the search arena dataset to questions with a single turn (not multi-turn), with a winner (no ties), and no longer than 5000 characters. We use a 2/3-1/3 train-test split.

---

### Official Review · Reviewer_Zm9C · 2025-07-02

**Clarity:** 2
**Significance:** 3
**Originality:** 3
**Rating:** 4
**Confidence:** 3

**Summary:**

This paper introduces Set-LLM, a permutation-invariant adaptation of decoder-only large language models (LLMs) designed to eliminate order sensitivity when processing mixed set-text inputs, such as multiple-choice options or sets of candidate responses. The authors provide theoretical proofs and comprehensive empirical results demonstrating that Set-LLM improves robustness to input permutations while maintaining or surpassing the performance of baseline models.

**Questions:**

- Table 1 and Table 3, which report performance across models, should be merged to facilitate clearer, side-by-side comparisons of model sizes and methods.

**Ethical Concerns:**

["NO or VERY MINOR ethics concerns only"]

**Final Justification:**

After the rebuttal, my main concerns about the task's generalization ability were resolved. I lean toward acceptance of this paper.

**Limitations:**

Yes

**Quality:**

2

**Strengths And Weaknesses:**

**Strengths:**

- The paper tackles a clear and impactful vulnerability in LLMs—order sensitivity—by introducing a novel architectural solution that guarantees permutation invariance.
-The authors provide formal proofs of permutation invariance, offering a solid theoretical foundation for the proposed architecture.
-Experiments across five models and four multiple-choice datasets validate the effectiveness and generalization of Set-LLM.

**Weaknesses:**

- Some technical sections (e.g., the formalization of set-token mappings in Lines 104–107) are notation-heavy and complex and may benefit from clearer exposition or illustrative examples.
- While the multiple-choice QA setting is well-justified, experiments focus solely on this task; broader generalization to other set-input scenarios (e.g., retrieval-augmented generation, ranking) is left untested.
- Including baselines that do not provide input choices (i.e., original dataset prompt) seems tangential to motivation and are not sufficiently analyzed in the experimental section.

---

> ### Author Rebuttal · Authors · 2025-07-31
>
> We sincerely thank Reviewer Zm9C for the positive feedback and valuable suggestions to improve the paper.
>
>
> > W1: Some technical sections (e.g., the formalization of set-token mappings in Lines 104–107) are notation-heavy and complex and may benefit from clearer exposition or illustrative examples.
>
>
> Thank you for raising this concern. We have tried to include figures and illustrative examples to help the reader, but we will certainly go through the paper again and see if we can improve clarity. Indeed, the set-token mappings sound much more complicated than they are, and we included Figure 3 for this reason. We will take a close look at whether we can simplify this formalization for the final version. We hope the intuition however is clear.
>
>
> > W2: While the multiple-choice QA setting is well-justified, experiments focus solely on this task; broader generalization to other set-input scenarios (e.g., retrieval-augmented generation, ranking) is left untested.
>
>
> We performed additional experiments on retrieval-augmented generation and multi-document summarization. We describe these and discuss the results below.
>
> ## Multi Document Summarization
>
> We conduct our evaluation on the MultiNews [1] dataset. We filter out inputs of length greater than 20000 characters to satisfy our memory constraints. We use the standard train-validation-test split. We compare the original model architecture with Set-LLM with single runs (no adversarial setting). Our initial results do not include hyperparameter tuning, we use the best-performing hyperparameters from the multiple-choice experiments. The experiments will be expanded for the final paper.
>
> We report standard metrics, this includes Rouge F1 scores and compression - the length of the original text divided by the length of the summary. We also include Rouge F1 scores between the target summaries and the model summaries (R1*, R2*, R3*).
>
> |Model             |Variant        |Compression |R1      |R2      |RL      |R1*      |R2*     |RL*    |
> |-------------------|----------------|-------------------|---------|---------|---------|---------|---------|---------|
> |Target             |                    |5.18               |0.30    |0.14    |0.16    | -        | -         | -        |
> |Gemma 2B     |Original       |13.21             |0.19    |0.14    |0.15    |0.32    |0.10    |0.19   |
> |                       |Set-LLM      |6.39               |0.30    |0.23    |0.23    |0.48    |0.19    |0.26   |
> |Gemma 7B     |Original       |8.11               |0.27    |0.21    |0.22    |0.42    |0.15    |0.23   |
> |                       |Set-LLM      |6.64               |0.30    |0.23    |0.23    |0.50    |0.22    |0.28   |
> |Llama 3.2 3B  |Original       |8.86               |0.25    |0.17    |0.18    |0.42    |0.14    |0.22   |
> |                       |Set-LLM      |6.31               |0.30    |0.20    |0.19    |0.50    |0.20    |0.26   |
>
> Set-LLM outperforms the finetuned baseline with all three base models in all metrics. However, the margin is significantly narrower with the larger Gemma 7B model. Note that the compression rate is always lower for Set-LLM, meaning the summaries it produces are longer.
>
> [1] Fabbri, Alexander R., et al. "Multi-news: A large-scale multi-document summarization dataset and abstractive hierarchical model." arXiv preprint arXiv:1906.01749 (2019).
>
>
> ## Multi-Document Question Answering (Retrieval-Augmented Generation)
>
> Motivated by [2], we evaluate Set-LLM on retrieval-augmented question answering. The authors introduce a task mimicking the retrieval-augmented generation setup. They ask questions with 20 supporting documents, where only one document contains the information required for the answer. They show that commercial LLMs (GPT3.5 Turbo) produce a U-shaped performance graph given the location of the answer. If the document with the answer is listed first or last, then the LLM answers the question correctly with a much higher probability (e.g. 75% in position 1/20 vs 55% in position 10/20). In contrast, Set-LLM is guaranteed to have a flat performance curve, since reordering the documents does not affect the outcome.
>
> We split the dataset into 60% training, 10% validation, and 30% test data. We replicate the training data 5 times and shuffle the documents randomly for each input. This is done to avoid the model overfitting to a specific formulation of each training question. We have not conducted hyperparameter tuning yet, the hyperparameters were taken from the multiple-choice experiments.
>
> We have 5 different test sets with the answer document placed in 5 different locations (first, 5th, 10th, 15th, last). We provide the results for two LLama-based models below. All scores denote the percentage accuracy.
>
> |Model             |Variant        |1         | 5        | 10     | 15      | 20      |
> |-------------------|----------------|---------|---------|---------|---------|---------|
> |Llama 3.2 1B  |Original      |34.76  |26.85  |25.22  |28.23  |37.39  |
> |                       |Set-LLM     |76.54  |76.54  |76.54  |76.54  |76.54  |
> |Llama 3.2 3B  |Original      |54.83  |44.29  |44.92  |44.04  |52.07  |
> |                       |Set-LLM     |81.68  |81.68  |81.68  |81.68  |81.68  |
>
> [2] Nelson F. Liu, Kevin Lin, John Hewitt, Ashwin Paranjape, Michele Bevilacqua, Fabio Petroni, and Percy Liang. 2024. Lost in the Middle: How Language Models Use Long Contexts. Transactions of the Association for Computational Linguistics, 12:157–173.
>
>
> > W3: Including baselines that do not provide input choices (i.e., original dataset prompt) seems tangential to motivation and are not sufficiently analyzed in the experimental section.
>
>
> This is a great suggestion. We have run experiments with a pretrained model and with a model (pre-)finetuned on the ultrafeedback dataset and then finetuned on the multiple choice datasets separately. We finetuned these models again, because the dataset now doesn't include the choices in the question. We report results below for Gemma 2B, but we will include results for all models in the final paper. There is no adversarial setting for this baseline. We include Set-LLM scores for comparison. We see that our model clearly outperforms this additional baseline.
>
>
> |Variant                           |PIQA   |ARC  |CSQA |SIQA  |
> |--------------------------------|---------|---------|---------|---------|
> |Causal Mask+PE           |76.77  |37.80  |51.76  |37.26 |
> |Causal Mask+PE Ultra  |79.87  |43.26  |69.37  |56.55 |
> |Set-LLM + Ultra             |85.80  |65.02  |80.18  |76.15 |

---

> > ### Comment · Reviewer_Zm9C · 2025-08-05
> >
> > Thanks the authors for all the clarifications and efforts. Most of my concerns are relatively well addressed. Please consider reflecting these results in the updated manuscript if possible.
> >
> > I will retain my score, which leans toward acceptance of this paper.

---

### Note · Authors · 2025-08-14

Dear Reviewers, ACs, and SACs,

We thank all reviewers for their valuable feedback and continued engagement throughout the discussion phase. We are glad that the reviewers acknowledge that “the paper tackles a clear and impactful vulnerability in LLMs” (Zm9C) and that the problem is “timely and important” (nn3i). Key points from the reviews and subsequent discussions are summarized as follows.

**Scope & Impact** – Based on feedback from multiple reviewers (Zm9C, ryD4, f4vf), we added experiments to showcase the strength of our approach in new real-world applications: retrieval-augmented generation (RAG) and multi-document summarization. After widening the experimental scope, there is now a general consensus about the significant impact of our paper.

**Strong Empirical Performance and Theoretical Guarantees** – We are happy that the reviewers appreciate the theoretical guarantees of our model, which are supported by strong empirical performance. We added a new baseline (thanks to Zm9C) and a new evaluation metric (thanks to f4vf) based on the discussions to further strengthen these results.

**Non Multiple-Choice Performance** – Two reviewers (ryD4, nn3i) asked about the impact of set-invariant finetuning on the general (non-multiple-choice) capabilities of the LLMs. We provided strong evidence that the general LLM abilities (translation and question-answering) are preserved, though the specific multiple-choice finetuning setup can lead to models that copy the input.

We believe our paper’s strengths and potential impact in the field, along with the planned additions, make this work a valuable submission.

---

### Decision · Program_Chairs · 2025-09-17

**Decision:**

Accept (poster)

**Comment:**

This paper addresses an important problem with transformers: that in certain settings, such as multiple-choice-question (MCQ) answering, the LLM's choice should not depend on the order in which the answer options are presented.  In practice, it does depend on the order, which reduces the utility of MCQ as a benchmark. This paper addresses the problem with a suite of technical innovations in the low-level transformer architecture to make "set permutation invariance" a first-class capability of LLMs, and demonstrate that the techniques do in fact improve permutation invariance.

While the authors had some initial concerns with the paper, especially regarding limited applicability / scope of the method, most of the concerns were resolved with a robust author-reviewer discussion. The reviewers are now satisfied with the paper. Given its general potential interest to the field, and given the reviewers' positive ratings, I recommend acceptance.